# Involution of brown adipose tissue through a Syntaxin 4 dependent pyroptosis pathway

Xiaofan Yu [1,2], Gabrielle Benitez [1,2], Peter Tszki Wei [3], Sofia V. Krylova [1,2,4], Ziyi Song [5], Li Liu [1,2], Meifan Zhang [6], Alus M. Xiaoli [1,2,7], Henna Wei [1], Fenfen Chen [8], Simone Sidoli [9], Fajun Yang [1,2,7], Kosaku Shinoda [1,2,4], Jeffrey E. Pessin [1,2,4] & Daorong Feng [1,2] ✉

Aging, chronic high-fat diet feeding, or housing at thermoneutrality induces brown adipose tissue (BAT) involution, a process characterized by reduction of BAT mass and function with increased lipid droplet size. Single nuclei RNA sequencing of aged mice identifies a specific brown adipocyte population of *Ucp1*-low cells that are pyroptotic and display a reduction in the longevity gene syntaxin 4 (*Stx4a*). Similar to aged brown adipocytes, Ucp1-STX4KO mice display loss of brown adipose tissue mass and thermogenic dysfunction concomitant with increased pyroptosis. Restoration of STX4 expression or suppression of pyroptosis activation protects against the decline in both mass and thermogenic activity in the aged and Ucp1-STX4KO mice. Mechanistically, STX4 deficiency reduces oxidative phosphorylation, glucose uptake, and glycolysis leading to reduced ATP levels, a known triggering signal for pyroptosis. Together, these data demonstrate an understanding of rapid brown adipocyte involution and that physiologic aging and thermogenic dysfunction result from pyroptotic signaling activation.

Adipose tissue plays a myriad of roles in the maintenance of normal physiology, and dysregulation of its functions can result in severe metabolic disorders. There are three general classes of adipocytes, referred to as white, brown and beige adipocytes[1–4]. White adipocytes primarily function as an energy reservoir by accumulating fatty acids in the form of triglycerides during periods of energy excess and releasing fatty acids during periods of energy deficiency[5,6]. In addition, white adipocytes play an important endocrine function through the secretion of a variety of adipokines and lipokines as mediators that have potent regulatory actions[7–9]. Brown and beige adipocytes also have important endocrine functions but primarily serve as thermogenic tissue by activating a variety of futile cycles resulting in substantial heat generation during periods of cold stress[10–14]. In rodents, the major difference between brown and beige adipocytes is that brown adipose tissue (BAT) depots are present throughout life while beige adipocytes are inducible and arise from within the white adipose tissue depots[3]. Humans are born with both white and brown adipose tissue, but the major BAT depot, interscapular brown adipose tissue (iBAT), disappears by the second to third decade of life[15–17]. Recent studies have shown that a subset of adult humans have brown-like adipose tissue in cervical, supraclavicular, axillary, paravertebral, and perirenal regions that are cold-inducible[18,19]. However, it remains debatable whether these human depots are the equivalent of brown or beige adipose tissue in rodents[20,21].

[1]Department of Medicine, Albert Einstein College of Medicine, Bronx, NY 10461, USA. [2]Fleischer Institute for Diabetes and Metabolism, Albert Einstein College of Medicine, Bronx, NY 10461, USA. [3]Department of Biological and Environmental Engineering, Cornell University, Ithaca, NY 14853, USA. [4]Department of Molecular Pharmacology, Albert Einstein College of Medicine, Bronx, NY 10461, USA. [5]Guangxi Key Laboratory of Animal Breeding, Disease Control and Prevention, College of Animal Science and Technology, Guangxi University, Nanning, Guangxi 530004, China. [6]Center for Advanced Biotechnology and Medicine, Rutgers University, Piscataway, NJ 08854, USA. [7]Department of Developmental and Molecular Biology, Albert Einstein College of Medicine, Bronx, NY 10461, USA. [8]Department of Animal Science, College of Life Science, Southwest Forestry University, Kunming, Yunnan 650244, China. [9]Department of Biochemistry, Albert Einstein College of Medicine, Bronx, NY 10461, USA. ✉e-mail: daorong.feng@einsteinmed.edu

In any case, classic mouse BAT undergoes an age-dependent degeneration process termed involution which is characterized by increased lipid deposition, increased size of lipid droplets, decreased mitochondria mass with reduction of thermogenic specific genes expression, and reduction in thermogenic activity[22–24]. This degenerative process can also be mimicked by maintaining mice at thermoneutrality ( ~ 28–30 °C) or by chronic high-fat diet feeding[21,25]. Since low BAT thermogenic activity is associated with poor metabolic health, reactivation and/or prevention of BAT involution has major therapeutic potential. However, the mechanisms responsible for the maladaptive BAT processes particularly during aging remain poorly understood.

Previous studies have shown that the whole-body overexpression of the SNARE (soluble N-ethylmaleimide-sensitive-factor attachment protein receptor) protein syntaxin 4 (STX4) increases longevity and improves body glucose metabolism[26]. Tissue-specific STX4 overexpression in beta cells improves insulin secretion and overexpression in skeletal muscle improves mitochondrial function[26,27]. We have found that aging-induced iBAT involution correlated with a reduction in STX4 expression, activation of pyroptotic signaling and loss of iBAT thermogenic activity. We reasoned that deficiency of STX4 in brown adipocytes should therefore impair thermogenic function whereas re-expression of STX4 protein in aged mice should preserve BAT function. We provide evidence that brown adipocyte STX4 deficiency mimics many of the involution characteristics that occur during aging, thermoneutrality and chronic high-fat diet feeding. More importantly, we identified a pyroptotic signaling pathway driven by Caspase1/11 activation, which is responsible for the loss of brown adipocyte mass and decrease in thermogenic activity probably through a dedifferentiation process generating *Ucp1*-low expressing brown adipocytes from *Ucp1*-high brown adipocytes. Moreover, tissue-specific STX4 overexpression in brown adipocytes or suppression of pyroptosis activation protects against the decline in both mass and thermogenic activity in the aged and Ucp1-STX4KO mice.

## Results

### Involution of brown adipocytes with aging is associated with the increase of pyroptotic *Ucp1*-low population

It is well established that during the normal aging process of mice, involution of iBAT results in a small decrease in mass with thermogenic dysfunction along with apparent "whitening" of the brown adipocytes through an increase in lipid droplet size and reduction in the number of small multiloculated lipid droplets[23,28]. To assess the cellular changes that occur during iBAT aging, we performed single-nuclei RNA-sequencing (snRNAseq) analyses using a modified version of our established protocol[29]. Figure 1a displays a combined UMAP plot of cell clusters from 2-, 12- and 24- month-old mouse iBAT, with the upper UMAP plots showing the presence of the adipocyte clusters, and the lower UMAP plots including the fibroblast, endothelial, and progenitor cell clusters. Additionally, there is a very small group of cells that display characteristics typical of both adipocytes and macrophages, that we termed indeterminate. In 2-month-old iBAT, there is one primary brown adipocyte cluster that can be defined by high levels of *Ucp1* (Fig. 1b). In 12-month-old iBAT there is a small decrease in this *Ucp1*-high adipocyte population associated with a small increase in a *Ucp1*-low expressing adipocyte population (Fig. 1b). While, as the mice further age to 24 months, there is a further decline in the *Ucp1*-high population and increase of *Ucp1*-low adipocyte cell population. The heat map in Fig. 1c shows the relative expression of several brown adipocyte markers in the two adipocyte clusters as a function of age.

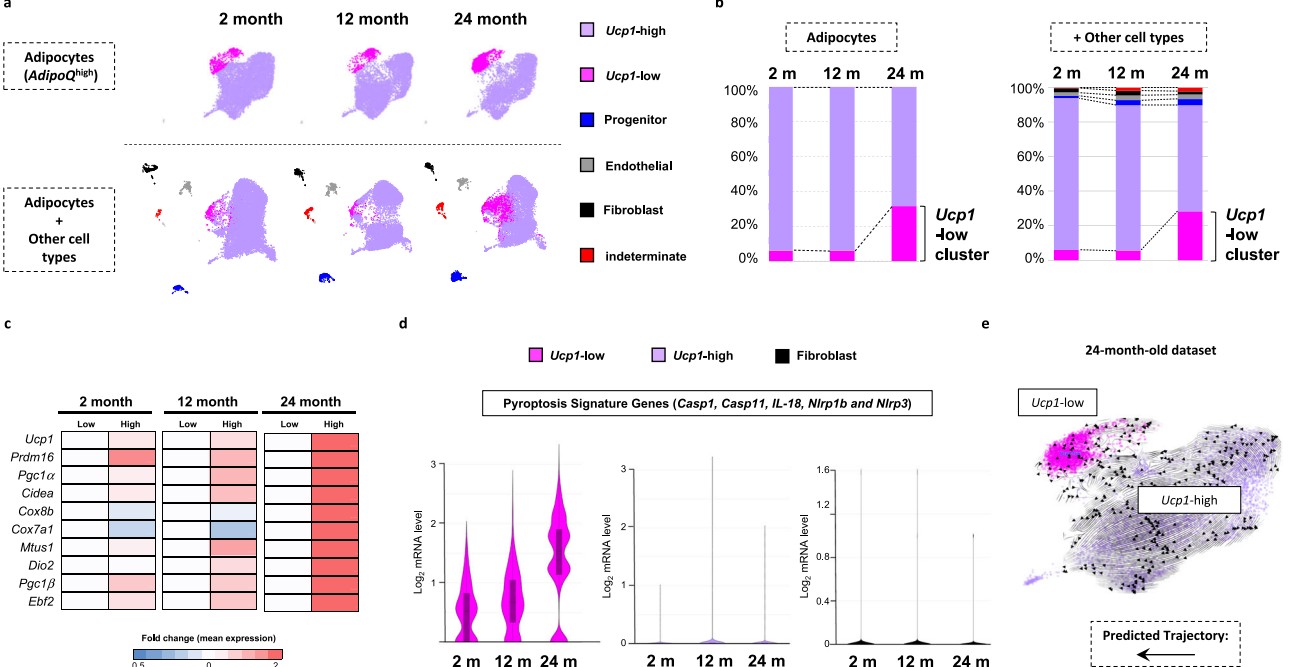

**Fig. 1 | The involution of brown adipocytes with aging is associated with the increase of pyroptotic *UCP1-low* population. a** Uniform manifold approximation and projection (UMAP) projection of all 27,678 sequenced nuclei displayed from iBAT from 2, 12 and 24-month-old male mice. The lower panel shows the same data set but with non-parenchymal stromal cells (30,000 nuclei in total). **b** The relative percent distribution of cell types present obtained in (**a**). Percentage (%) normalized to the number of total nuclei after exclusion of low-quality nuclei (see Methods). **c** Heatmap of the relative expression level of brown adipocyte thermogenic and marker genes for each cellular subtypes at 2, 12, and 24 months of age. **d** Violin plots showing age-related changes in the expression levels of pyroptotic gene markers in the *Ucp1*-low, *Ucp1*-high brown adipocytes and fibroblast populations. The boxplots show median and interquartile range (IQR); the lower and upper bounds correspond to the first and third quartiles. The upper whisker extends from the hinge to the largest value no further than 1.5*IQR from the hinge. The lower whisker extends from the hinge to the smallest value at most 1.5*IQR from the hinge. **e** Lineage relationships between cellular subtypes predicted by the RNA velocity algorithm 30. Arrows represent inferred cell interconversions. *n* = 4 mice.

The *Ucp1*-high adipocytes also display high levels of brown adipocyte specific markers, which are substantially reduced in the *Ucp1*-low adipocyte population (Fig.1c). Pyroptosis is a specific form of inflammatory programmed cell death that results from the assembly of inflammasome complex that stimulates Caspase1 and/or 11 activation, which in turn cleaves Gasdermin D, interleukin 1b (IL-1β) and interleukin 18 (IL-18)[30–32]. Violin plots demonstrated that both the aged (12 and 24 months) brown adipocytes display higher levels of pyroptotic gene markers than 2-month-old brown adipocytes in the *Ucp1*-low population (Fig. 1d). In contrast, there is no significant effect of age on the expression of pyroptotic gene markers in the *Ucp1*-high population and fibroblast cell population (Fig. 1d). RNA velocity analyses by inferred trajectories[33,34] indicate that there is an apparent cellular interconversion between the *Ucp1*-low and *Ucp1*-high adipocyte populations (Fig. 1e). In addition, a subpopulation of the *Ucp1*-low adipocytes appears to have a de-differentiating trajectory.

To confirm this data, we isolated iBAT brown adipocytes (BA) and stromal vascular fraction (SVF) from 2- and 18-month-old mice. The expression of the adipocyte marker adiponectin (*Adipoq*) was 30 to 50-fold higher in brown adipocytes compared to SVF (Supplementary Fig. 1a). In contrast, the expression of *Adgre1* (macrophage marker) was 10 to 17-fold higher in the SVF than in the brown adipocytes (Supplementary Fig. 1b), confirming the successful separation of the two fractions. The expression levels of *Ucp1* and adiponectin were decreased while pyroptotic genes *Casp1, Casp11, Nlrp3, Il18* and *Il1b* were increased in the brown adipocytes from 18month-old mice (Supplementary Fig. 1c). In the SVF fraction, the 18-month-old mice had increased expression of *Adgre1, Casp1, Nlrp1b*, and *Il1b* (Supplementary Fig. 1d). These data indicate that both the brown adipocytes and stromal vascular cells within brown adipose tissue display induction of pyroptotic gene expression in aged mice. Consistent with morphological iBAT involution, phase contrast microscopy demonstrated an increase in lipid droplet size in the iBAT of aged mice along with decreased UCP1 labeling in various regions within the tissue (Supplementary Fig. 1e).

## Overexpression of STX4 in brown adipocytes protects against iBAT involution

In our snRNAseq data, the average expression level of *Stx4a* is 0.246 UMI count for *Ucp1*-high and 0.073 for *Ucp1*-low. Consistently, the *Stx4a* mRNA levels were also decreased in the isolated brown adipocytes from the aged male mice (Supplementary Fig. 1c). Moreover, immunoblotting demonstrated the downregulation of STX4 protein in the 12- and 24-month-old male mice compared to 2-month-old male mice (Supplementary Fig. 1f, g). Moreover, there was also a clear decline in both STX4 protein and UCP1 protein in 17-month-old female mice compared to 1-month-old female mice (Supplementary Fig. 1h, I, j) consistent with the findings in male mice. Previous studies have reported that STX4 can function as a healthy longevity gene[26]. If the reduction in STX4 expression is an important event in the involution of BAT, then maintaining normal levels of STX4 should prevent the functional decline of brown adipocytes. To address this, we inserted a cassette consisting of loxP-tPA (triple polyA signal)-loxP-mouse *Stx4a* cDNA into intron 1 of Rosa26 locus by CRISPR/HDR strategy, then crossed this transgenic line with Ucp1-Cre mice to generate a brown adipocyte-specific STX4 transgenic mice (Ucp1-STX4TG or STX4-TG mice) (Supplementary Fig. 2a). Expression of a single copy of the STX4-TG resulted in an approximate 2-fold increase in *Stx4a* mRNA and an approximate 1.5-fold increase in STX4 protein (Supplementary Fig. 2b–d), but with no change in *Stx4a* mRNA in the SVF (Supplementary Fig. 2b).

To induce BAT involution, 4-month-old control wild-type (WT) and STX4-TG mice were maintained at thermoneutrality for 1, 2 and 18 weeks and then directly shifted from 29 °C to 4 °C (Fig. 2a). The STX4-TG mice displayed a significant protection against hypothermia compared to the control wild-type mice after 1, 2 and 18 weeks of housing at thermoneutrality (Fig. 2b–d). Although the expression of adipocyte gene marker (*Adipoq*) was unchanged in the STX4-TG mice, the expression levels of Caspase1, 11 and *Il1b* were reduced (Fig. 2e). Despite no change in *Ucp1* mRNA level, UCP1 protein levels were increased in the iBAT from the STX4-TG mice along with reduced NLRP1b cleavage (activation) protein levels (Fig. 2f–h). In parallel, plasma IL-1β levels were reduced (Fig. 2i), and a protection against the morphological appearance of whitening with the presence of brown adipocytes containing smaller multi-locular lipid droplets in the STX4-TG mice (Fig. 2j). The STX4-TG mice also displayed improved insulin sensitivity compared to control wild-type mice (Fig. 2k). To examine any potential sexual dimorphism, we compared the effect of the STX4-TG mice at thermoneutral conditions between male and female mice (Supplementary Fig. 2e–j). Both male and female STX4-TG mice displayed reduced body mass and fat mass gain with no change in lean mass. We also found that there was a significant reduction in scWAT weight of WT male mice following 8 months under thermoneutral conditions (Supplementary Fig. 2l). In addition, the female STX4-TG mice also displayed protection against cold-induced hyperthermia similarly to that observed for the male mice (Supplementary Fig. 2k). To determine the effect of STX4-TG on aged-induced involution, 17-month-old STX4-TG female mice BAT tissue demonstrated an increase in both UCP1 mRNA and protein levels (Supplementary Fig. 2m–o).

## Knockout of STX4 in brown adipocytes leads to age-dependent involution of brown adipose tissue and activation of pyroptosis

To further confirm the function of STX4 in brown adipose tissue, we then generated brown adipocyte-specific STX4-deficient mice by crossing floxed STX4 mice with Ucp1-Cre mice to generate Ucp1-Cre[+/-]STX4[fl/fl] (Ucp1- STX4KO or KO) mice. Quantitative real-time PCR (RT-qPCR) demonstrated an approximate 90% reduction in *Stx4a* mRNA from the purified brown adipocytes of Ucp1- STX4KO mice compared with control Ucp1-Cre[+/-] or STX4[fl/fl] mice (WT), but no change in *Stx4a* mRNA from the purified stromal vascular fraction (SVF) compared to WT mice (Supplementary Fig. 3a).

There were no significant differences in body weight of Ucp1-STX4KO mice when compared to their wild-type (WT) littermates across the age range of 2–9 months and the data for 5-month-old male mice as shown in Supplementary Fig. 3b. The weight of the subcutaneous white adipose tissue (scWAT) in the Ucp1- STX4KO mice was also not significantly different compared to wild-type (WT) littermates at 5 months of age (Supplementary Fig. 3c). Although the Ucp1- STX4KO mice visually appeared normal, dissection revealed a marked decline in interscapular brown adipose tissue (iBAT), without any significant effect on white adipose tissue (Fig. 3a). The reduction in iBAT mass occurred as a function of age with near complete ablation by 9 months of age (Fig. 3b). Despite only an approximate 40% reduction of iBAT in the 2-month-old mice, they displayed a marked increase in cold-induced hyperthermia compared to the control wild-type (WT) mice (Fig. 3c).

To investigate the molecular basis for the loss of iBAT mass, we perform bulk RNA sequencing of iBAT at 2 months of age. The heatmap shown in Supplementary Fig. 3d displays the differentially expressed genes between WT and Ucp1-STX4KO mice (*n* = 4 for each group). KEGG pathway enrichment analysis indicated that thermogenesis is the most significantly down-regulated pathway in the Ucp1-STX4KO mice shown in the reduction of several pathways including oxidative phosphorylation, branched amino acid metabolism, and fatty acid degradation (Supplementary Fig. 3e), all of which are linked to brown fat thermogenesis[35]. In contrast, pyroptosis, inflammatory signaling, and oxidative stress pathways were increased in the Ucp1-STX4KO mice (Supplementary Fig. 3f, g). Further analyses of the RNA sequencing data demonstrated that many of the pyroptotic signaling gene family members were upregulated in the iBAT of the UCP1-STX4KO mice (Fig. 3d). Quantitative real-time PCR (RT-qPCR) for

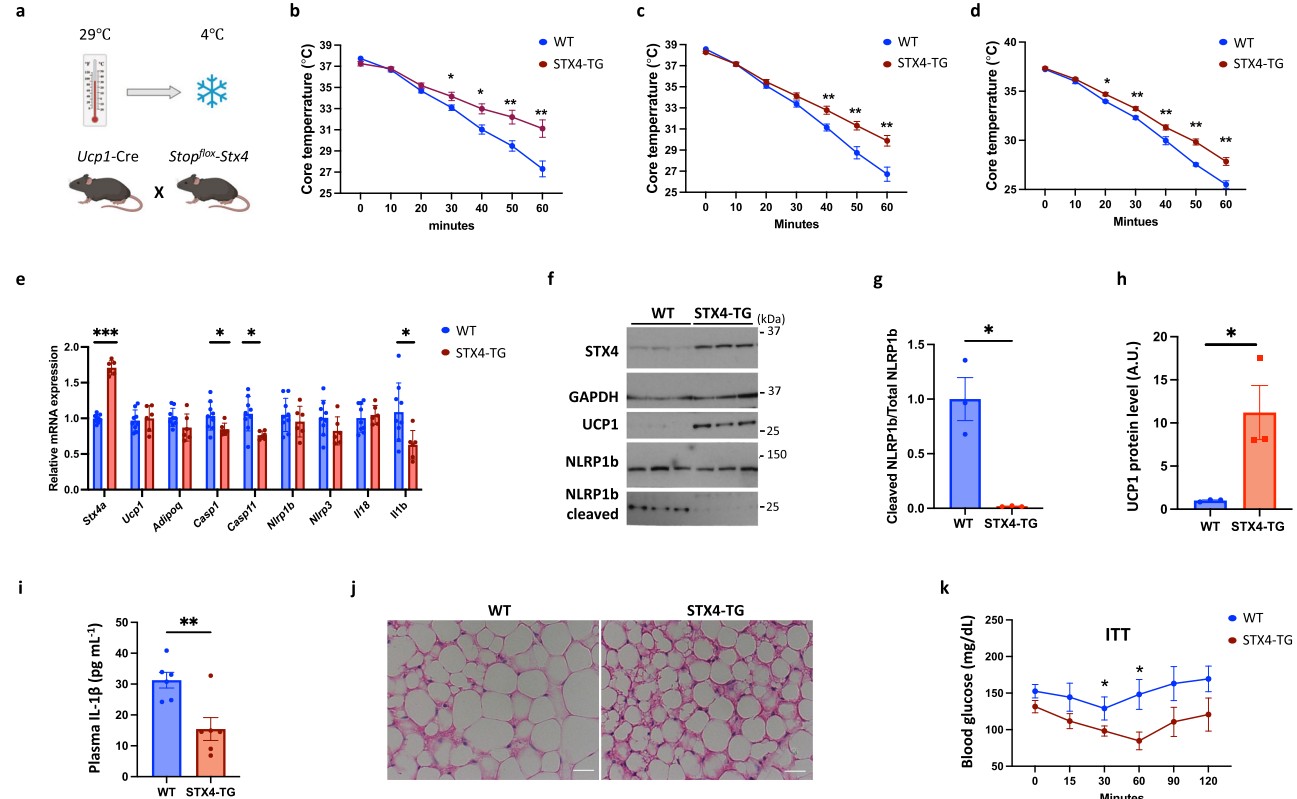

**Fig. 2 | Overexpression of STX4 in brown adipocytes protects against iBAT involution. a** Schematic diagram of experiment setup. **b** Core temperature of wild-type (WT) and brown adipocyte specific transgenic STX4-TG male mice maintained at thermoneutrality (29 °C) for 1 week then shifted to 0 °C for 60 min. $n = 12$ (WT), $n = 6$ (STX4-TG) mice. $p = 0.034$ (30 min), $p = 0.013$ (40 min), $p = 0.005$ (50 min), $p = 0.006$ (60 min). **c** Core body temperature of WT and STX4-TG male mice maintained at thermoneutrality 2 weeks then acutely cold challenged as in (**b**). $n = 12$ (WT), $n = 6$ (STX4-TG) mice. $p = 0.007$ (40 min), $p = 0.009$ (50 min), $p = 0.007$ (60 min). **d** Core body temperature of WT and STX4-TG male mice maintained at thermoneutrality 18 weeks then acutely cold challenged as in (**b**). $n = 3$ (WT), $n = 6$ (STX4-TG) mice. $p = 0.021$ (20 min), $p = 0.014$ (30 min), $p = 0.01$ (40 min), $p = 0.001$ (50 min), $P = 0.002$ (60 min). **e** Brown adipose tissue from WT and STX4-TG male mice maintained at thermoneutrality for 2 weeks was extracted and subjected to RT-qPCR to determine the indicated mRNA levels. $n = 9$ (WT), $n = 6$ (STX4-TG) mice.

$p = 1.7E{-}10$ (*Stx4a*), $p = 0.05$ (*Casp1*), $p = 0.01$(*Casp11*), $p = 0.025$ (*Il-1b*). **f** Immunoblot for STX4, UCP1, NLRP1b and cleaved NLRP1b in brown adipose tissue of WT and STX4-TG male mice at thermoneutrality for 2 weeks. **g** Quantitation of cleaved NLRP1b to total NLRP1b in WT and STX4-TG male mice maintained at thermoneutrality for 2 weeks. $n = 3$ mice. $p = 0.038$. **h** Quantitation of UCP1 protein level in WT and STX4-TG male mice maintained at thermoneutrality for 2 weeks. $n = 3$ mice. $p = 0.046$. **i** IL-1β in plasma of WT and STX4-TG male mice maintained at thermoneutrality for 2 weeks. $n = 5$ mice. $p = 0.006$. **j** HE staining of brown adipose tissue of WT and STX4-TG male mice maintained at thermoneutrality for 18 weeks. Scale bars: 100 μM. **k** ITT of WT and STX4-TG male mice maintained at thermoneutrality for 18 weeks. $n = 5$ (WT), $n = 6$ (STX4-TG) mice. $p = 0.05$ (30 min). $p = 0.021$ (40 min). All data represent the mean ± SEM. *$p < 0.05$, **$p < 0.01$, and ***$p < 0.005$ by two-tailed Student's *t*-test.

several of these gene markers confirmed a 2- to 8-fold elevation in mRNA in the iBAT of the Ucp1-STX4KO mice (Fig. 3e), resembling pyroptosis activation in aged BAT shown in Fig. 1.

Immunohistochemistry demonstrated the presence of cleaved (activated) Caspase1, and quantification indicated a 3-fold increase in iBAT of the Ucp1-STX4KO mice (Fig. 3f, g). Consistent with these data, immunoblotting demonstrated a decrease in the total amount of full-length Caspase1 with a concomitant increase in cleaved Caspase1 (Fig. 3h). Furthermore, there was a marked induction of the Nod-like receptor NLRP1b protein in the Ucp1-STX4KO brown adipose tissue, but without any significant change in NLRP3 or AIM2 proteins (Fig. 3i). Since pyroptosis is an inflammatory cell death process, we also observed a clear loss of brown adipocytes as detected by the reduction in perilipin staining and increased presence of macrophages in the iBAT of the Ucp1-STX4KO mice (Fig. 3j, k).

To dissect the cellular heterogeneity of Ucp1-STX4KO at a single-cell resolution, we conducted snRNAseq on brown adipocytes derived from both WT and Ucp1-STX4KO mice. Our analysis revealed the presence of one major *Ucp1*-high and *Ucp1*-low adipocyte cell cluster (Supplementary Fig. 4a). *Ucp1 mRNA expression is significantly higher in Ucp1*-high *compared to Ucp1*-low (Supplementary Fig. 4b). Notably,

in the Ucp1-STX4KO mice there was a marked increase in the *Ucp1*-low adipocyte clusters with a parallel reduction in the *Ucp1*-high adipocyte clusters compared to WT mice (Supplementary Fig. 4c). The heat map of pyroptotic and adipocyte marker genes displayed a similar pattern to the aged iBAT, that is the *Ucp1*-low cluster expressed higher levels of pyroptotic gene markers whereas the *Ucp1*-high clusters displayed higher levels of adipocyte marker gene (Supplementary Fig. 4d). In addition, RNA velocity analyses were suggestive of interconversion between the *Ucp1*-high and *Ucp1*-low adipocyte populations as well as of a portion of the *Ucp1*-low adipocytes undergoing de-differentiation (Supplementary Fig. 4e). The reduction in UCP1 protein in a subset of the brown adipocytes is also readily visualized by immunofluorescence (Supplementary Fig. 4f). Together, these data support the hypothesis that STX4 deficiency mimics, at least in part, the aged-induced involution of brown adipose tissue.

### Inhibition of Caspase 1/11 mediated-pyroptosis prevents involution of brown adipose tissue in STX4 knockout mice

As Caspase1/11 is the downstream effector of pyroptosis signaling, we next determined the in vivo consequence of pharmacological Caspase1/11 inhibition. Littermate control (WT) and Ucp1-STX4KO mice

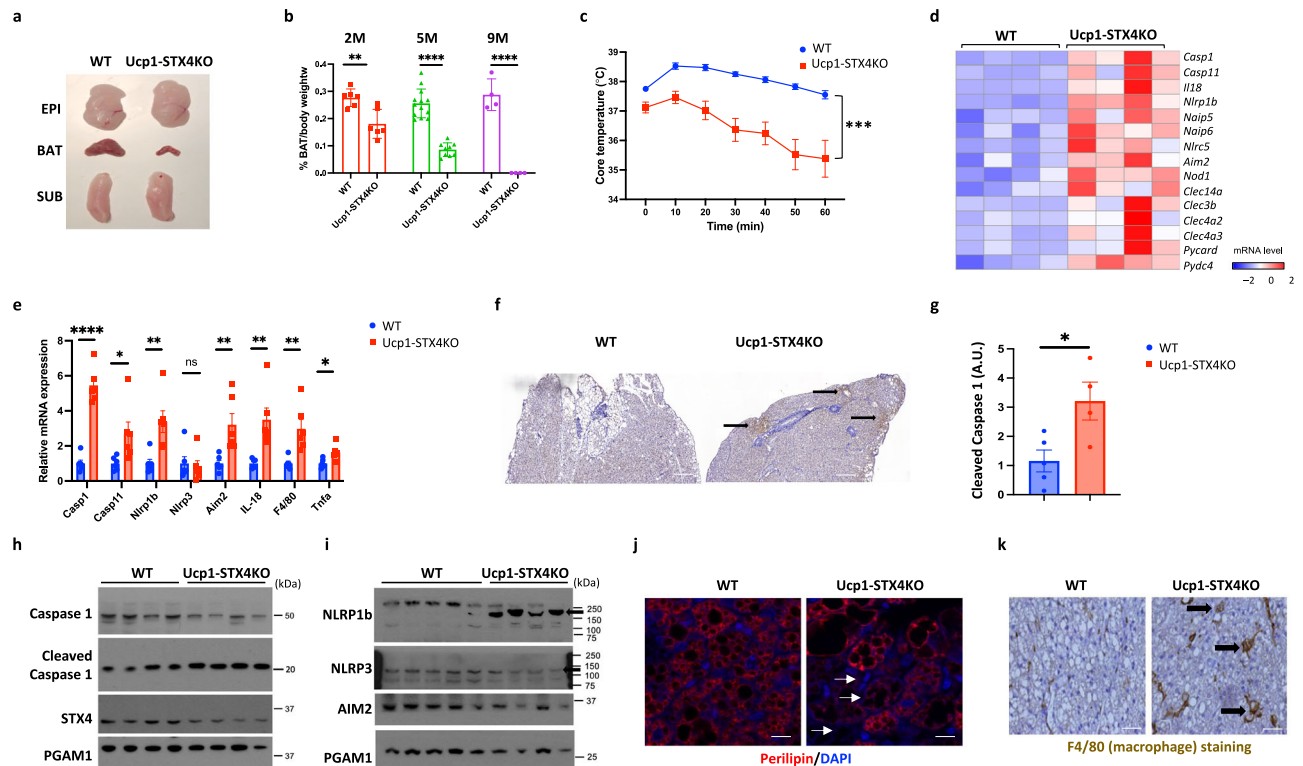

**Fig. 3 | Knockout of STX4 in brown adipocytes leads to age-dependent involution of brown adipose tissue and activation of pyroptosis. a** The representative image of different fat pad from 5.5 - month - old WT and Ucp1-STX4KO male mice. EPI, epididymal white adipose tissue; BAT, interscapular brown adipose tissue; SUB, subcutaneous white adipose tissue. **b** Brown adipose tissue mass in WT and Ucp1-STX4KO male mice at 2 months (2M), 5 months (5M), and 9 months (9M) of age. For WT mice, $n = 6$ (2M), $n = 13$ (5M), $n = 4$ (9M). For Ucp1-STX4KO mice, $n = 6$ (2M), $n = 9$ (5M), $n = 4$ (9M). $p = 0.0036$ (2M), $p = 2.01E{-}08$ (5M), $p = 6.1E{-}05$ (9M). **c** Core body temperature of WT and Ucp1-STX4KO male mice (2 months of age) maintained at room temperature and then shifted to 4 °C for 60 min. $n = 8$ (WT), $n = 5$ (Ucp1-STX4KO) *mice*. $p = 0.006$ (0 min), $p = 0.0004$ (10 min), $p = 0.0003$ (20 min), $p = 8.6E{-}05$ (30 min), $p = 0.0001$ (40 min), $p = 0.0002$ (50 min), $p = 0.0014$ (60 min). **d** The expression of pyroptotic genes in brown adipose tissue of Ucp1-STX4KO and WT male mice by RNAseq. $n = 4$ mice. **e** Brown adipose tissue from 2-month - old WT and Ucp1-STX4KO male mice was extracted and subjected to RT-

qPCR to determine the indicated mRNA levels. $n = 6$ mice. $p = 1.68E{-}06$ (*Casp1*), $p = 0.03$ (*Casp11*), $p = 0.005$ (*Nlrp1b*), $p = 0.725$ (*Nlrp3*), $p = 0.007$ (*Aim2*), $p = 0.004$ (*Il18*), $p = 0.004$ (*Adgre1*), $p = 0.011$ (*Tnfa*). **f** The activity of cleaved Caspase1 in BAT of Ucp1-STX4KO and WT male mice by immunostaining of cleaved Caspase1 antibody. **g** Quantitated signal of cleaved Caspase1 by HistoQuant. $n = 5$ (WT), $n = 4$ (Ucp1-STX4KO) mice. $p = 0.024$. **h, i** Representative immunoblotting of brown adipose tissue from 2- month - old WT and Ucp1-STX4KO male mice with different antibodies. PGAM1 as loading control. **j** Perilipin immunofluorescence (red) and DAPI (blue) staining of nuclei in brown adipose tissue of WT and Ucp1-STX4KO male mice at 2 months of age. Arrows indicate perilipin-depleted cells. Scale bars: 100 μM. **k** F4/80 immunostaining in brown adipose tissue in WT and Ucp1-STX4KO male mice at 2 months of age. Scale bars: 200 μM. All data represent the mean ± SEM. *$p < 0.05$, **$p < 0.01$, ***$p < 0.001$ and ****$p < 0.0001$ by two-tailed Student's $t$ test.

were injected with vehicle or selective Caspase1/11 inhibitor VX-765 compound for 2 months, 3 times per week, starting at 6 weeks of age. The vehicle-treated Ucp1-STX4KO mice displayed a reduction in iBAT mass compared to control mice whereas the VX-765-treated Ucp1-STX4KO mice had a smaller reduction in iBAT mass (Supplementary Fig. 5a). In parallel, the VX-765-treated Ucp1-STX4KO mice were also significantly protected against acute cold challenge induced hypothermia compared to the vehicle-treated Ucp1-STX4KO mice (Supplementary Fig. 5b).

Since this experimental paradigm resulted in only a partial protection against the loss of iBAT mass and cold-induced hypothermia, we speculated that brown adipocytes were already undergoing pyroptotic cell death by the time we started VX-765 treatment. Therefore, we next took advantage of inducible Ucp1-Cre^ERT2 mice crossed with the STX4^fl/fl mice in which we could pharmacologically treat the mice right after inducing STX4 deficiency (Fig. 4a). Using this approach there was an effective loss of STX4 protein in the inducible Ucp1-STX4KO (Ind-KO) mice whether they were treated with vehicle or VX-765 (Fig. 4b). As shown in Fig. 4b, VX-765 reduced the cleavage of Caspase1. But since VX-765 is an inhibitor of Caspase1/11 enzymatic activity this intervention should not affect upstream signaling events.

Analyses of the upstream pyroptotic signaling pathway demonstrated that VX-765 treatment did not suppress the expression of Caspase1 per se, or *Nlrp1b* mRNA but did block the increase in Il18 and Il1b as well as macrophage infiltration (Fig. 4c). Although VX-765 did not affect total body weight (Fig. 4d) and subcutaneous white adipose tissue mass (Fig. 4e), there was protection against the loss of iBAT mass (Fig. 4f). In line with this, VX-765 almost completely protected Ind-KO from acute cold-induced hypothermia (Fig. 4g). Importantly, the Caspase3 inhibitor (Z-DEVD-FMK) was not protective against the activation of Caspase1 nor the reduction of iBAT mass (Fig. 4h–k). Taken together these data support a model in which STX4 deficiency induces brown adipocyte cell death through a pyroptosis signaling pathway leading to the activation of the key effector caspases 1 and/or 11.

## Inhibition of Caspase1/11 mediated-pyroptosis prevents the involution of brown adipose tissue with aging

Since STX4 deficiency appears to mimic the aging process of iBAT, albeit in an accelerated fashion, and single nuclei RNAseq is non-quantitative, we next determined pyroptosis gene expression markers by quantitative RT-PCR in iBAT from 2-, 12- and 24-month-old mice (Fig. 5a). Several pyroptotic mRNAs (*Casp1, Aim2, Nlrp1b, Il18, Adgre1*

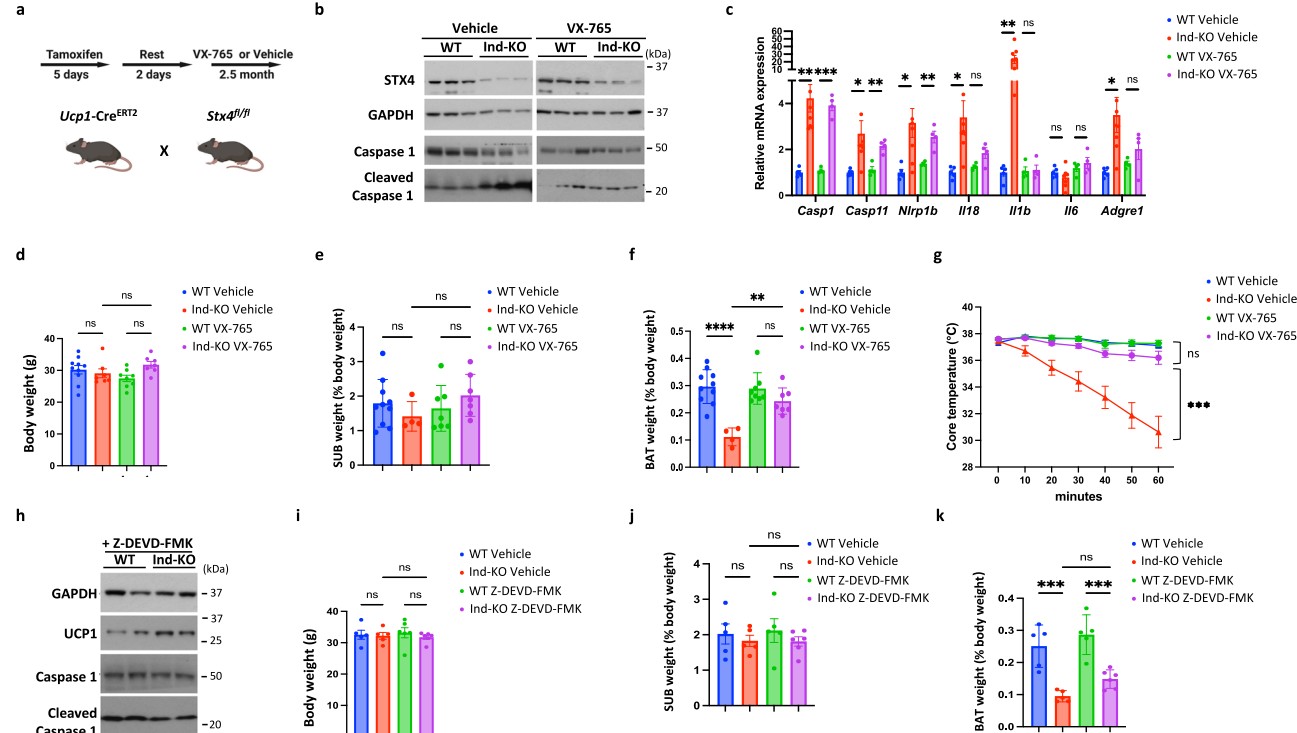

**Fig. 4 | Inhibition of Caspase 1/11 mediated-pyroptosis prevents the involution of brown adipose tissue in brown adipocyte specific STX4 knockout mice.**
**a** Schematic diagram of experiment setup. **b** Immunoblot of Syntaxin 4, Caspase1, cleaved Caspase1 and GAPDH from brown adipose tissue of wild-type (WT) and adipocyte-specific inducible STX4KO (Ind-KO) male mice treated with VX-765 (50 mg per kg of their body weight) or vehicle, three times a week for 2 months. $n = 3$ mice. **c** mRNA expression of pyroptotic genes in BAT of WT and Ind-KO male mice treated with VX-765 or vehicle by RT-qPCR. $n = 5$ (WT Vehicle), $n = 8$ (Ind-KO Vehicle), $n = 4$ (WT VX-765), $n = 4$ (Ind-KO VX-765) mice. Comparison of WT Vehicle with Ind-KO Vehicle, $p = 0.0015$ (*Casp1*), $p = 0.041$ (*Casp11*), $p = 0.023$ (*Nlrp1b*), $p = 0.0026$ (*Il18*), $p = 0.0072$ (*Il1b*), $p = 0.2975$ (*Il6*), $p = 0.027$ (*Adgre1*). Comparison of WT VX-765 with Ind-KO VX-765, $p = 2.06E-05$ (*Casp1*), $p = 0.0018$ (*Casp11*), $p = 0.0039$ (*Nlrp1b*), $p = 0.052$ (*Il18*), $p = 0.8669$ (*Il1b*), $p = 0.4474$ (*Il6*), $p = 0.2291$ (*Adgre1*). **d** Body weight of WT and Ind-KO male mice treated with vehicle or VX-765. $n = 10$ (WT Vehicle), $n = 7$ (Ind-KO Vehicle), $n = 8$ (WT VX-765), $n = 4$ (Ind-KO VX-765) mice. **e** Subcutaneous adipose tissue mass normalized to body weight in WT and Ind-KO male mice treated with vehicle or VX-765. $n = 10$ (WT Vehicle), $n = 4$ (Ind-KO Vehicle), $n = 7$ (WT VX-765), $n = 7$ (Ind-KO VX-765) mice. **f** Brown adipose tissue mass normalized to body weight in WT and Ind-KO male mice treated with vehicle or VX-765. $n = 10$ (WT Vehicle), $n = 4$ (Ind-KO Vehicle), $n = 8$ (WT VX-765),

$n = 7$ (Ind-KO VX-765) mice. $p < 0.0001$ (WT Vehicle *vs.* Ind-KO Vehicle), $p = 0.0041$ (WT VX-765 *vs.* Ind-KO VX-765). **g** Core body temperature of WT and Ind-KO male mice treated with VX-765 or vehicle following acute cold challenge from room temperature to 4 °C. $n = 10$ (WT Vehicle), $n = 8$ (Ind-KO Vehicle). $n = 8$ (WT VX-765). $n = 8$ (Ind-KO VX-765) mice. Comparison of WT Vehicle with KO Vehicle, $p = 0.0003$ (20 min), $p < 0.0001$ (30 min), $p = <0.0001$ (40 min), $p < 0.0001$ (50 min), $< 0.0001$ (60 min). **h** Western blot of GAPDH, UCP1, Caspase1 and cleaved Caspase1 from brown adipose tissue of WT and Ind-KO male mice treated with vehicle or Z-DEVD-EMK (2.5 mg per kg of their body weight, three times a week). $n = 3$ mice. **i** Body weight of WT and Ind-KO male mice treated with vehicle or Z-DEVD-FMK. $n = 5$ (WT Vehicle, WT VX-765 and Ind-KO Vehicle), $n = 6$ (Ind-KO VX-765) mice. **j** Subcutaneous adipose tissue mass normalized to body weight in WT and Ind-KO male mice treated with vehicle or Z-DEVD-FMK. $n = 5$ (WT Vehicle, WT VX-765 and Ind-KO Vehicle), $n = 6$ (Ind-KO VX-765) mice. **k** Brown adipose tissue mass normalized to body weight in WT and Ind-KO male mice treated with vehicle or Z-DEVD-FMK. $n = 5$ (WT Vehicle, WT VX-765 and Ind-KO Vehicle), $n = 6$ (Ind-KO VX-765) mice. $p = 0.0004$ (WT Vehicle *vs.* Ind-KO Vehicle), $p = 0.0008$ (WT DEVD *vs.* Ind-KO DEVD). All data represent the mean ± SEM. \**p* < 0.05, \*\**p* < 0.01 and \*\*\**p* < 0.001, by two-tailed Student's *t* test (**c**) or one-way ANOVA (**d**–**g**, **i**–**k**).

and *Tnfa*) were significantly increased in iBAT at both 12- and 24-month old mice compared to 2- month old mice concomitant with a reduction in both *Ucp1* and *Stx4a* mRNA (Fig. 5a). Immunoblotting confirmed the downregulation of UCP1 and STX4 proteins along with the upregulation of NLRP1b protein in the 12- and 24-month-old mice (Fig. 5b, Supplementary Fig. 1f, g). Quantification of NLRP1b and UCP1 protein levels are shown in Fig. 5c, d. We also observed a loss of brown adipocytes as detected by the reduction in perilipin staining in 24-month-old mice (Fig. 5e, f). Collectively, these data demonstrate that the activation of pyroptosis observed in the brown adipocyte specific STX4 deficiency model also occurs during physiological aging.

To determine if pharmacological inhibition of Caspase1/11 can also prevent age-associated iBAT involution, 8-month-old mice were placed under thermoneutrality to induce whitening and were injected with VX-765 or Z-DEVD-FMK (a selective Caspase3 inhibitor) for 4 months. VX-765 treatment reduced the plasma levels of IL-18 compared to vehicle or Z-DEVD-FMK-treated mice (Fig. 5g). Treatment with

VX-765 or Z-DEVD-FMK had no effect on body weight or subcutaneous white adipose tissue mass (Fig. 5h, i), but VX-765-treated mice displayed a small increase in the amount of iBAT mass (Fig. 5j). In parallel, the VX-765-treated mice also display greater protection against hypothermia compared to vehicle or Z-DEVD-FMK-treated mice (Fig. 5k). These data suggests that the loss of iBAT mass and thermogenic function during aging is also, at least in part, mediated by a Caspase1/11 activated pyroptosis.

## Knockout of STX4 causes Caspase 1/11-mediated pyroptosis in a cell-autonomous manner

We next accessed whether STX4 deficiency induced pyroptosis activation and subsequent cell death in a cell-autonomous manner. To accomplish this, we isolated brown adipocytes progenitors from the STX4[fl/fl] mice and then differentiated them into brown adipocytes. Brown adipocytes were differentiated for 6 days and then were infected with either an adenovirus expressing Cre (Ade-Cre) or a

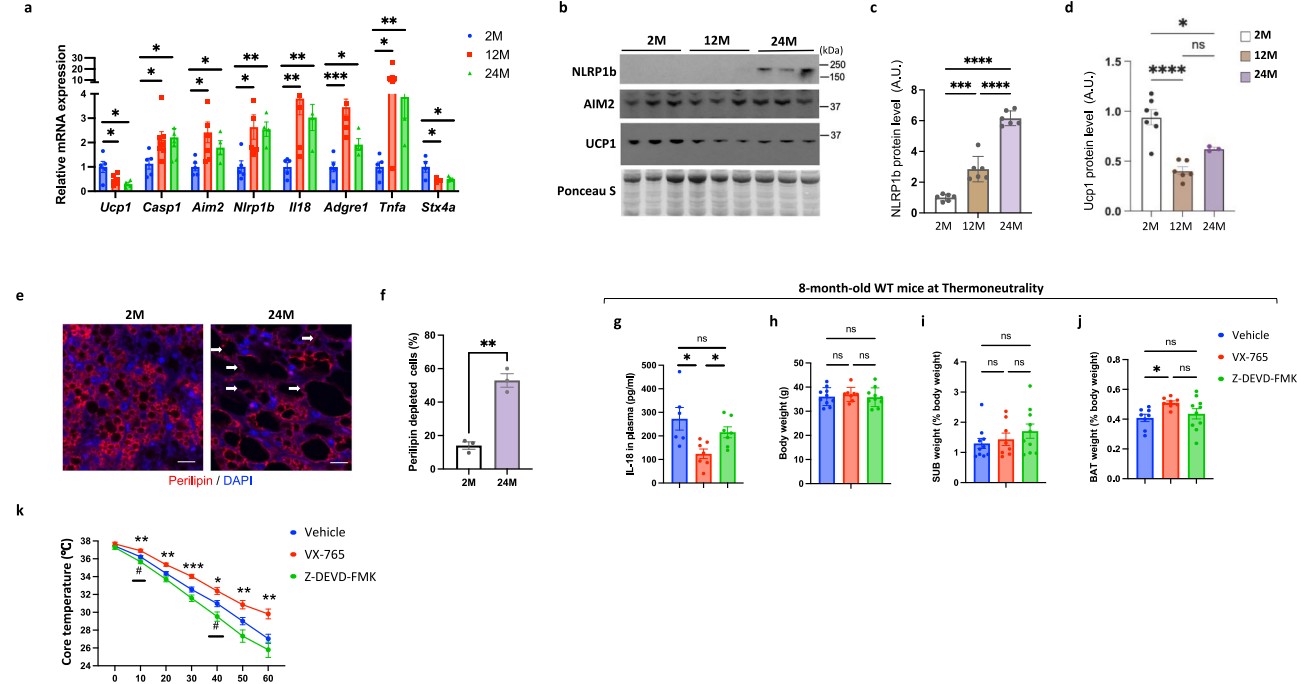

**Fig. 5 | Inhibition of Caspase1/11 mediated-pyroptosis prevents the involution of brown adipose tissue with aging. a** Brown adipose tissue from WT male mice at 2 months (2M), 12 months (12M), and 24 months (24M) of age was extracted and subjected to RT-qPCR to determine the indicated mRNA levels. n = 5 mice. Comparison of 2M with 12M, $p = 0.023$ (*Ucp1*), $p = 0.042$ (*Casp1*), $p = 0.023$ (*Aim2*), $p = 0.027$ (*Nlrp1b*), $p = 0.006$ (*Il8*), $p = 0.0002$ (*Adgre1*), $p = 0.017$ (*Tnfa*), $p = 0.042$ (*Stx4a*). Comparison of 2M with 24M, $p = 0.031$ (*Ucp1*), $p = 0.047$ (*Casp1*), $p = 0.035$ (*Aim2*), $p = 0.005$ (*Nlrp1b*), $p = 0.005$ (*Il8*), $p = 0.021$ (*Adgre1*), $p = 0.0086$ (*Tnfa*), $p = 0.037$ (*Stx4a*). **b** Immunoblots of proteins NLRP1b, AIM2 and UCP1 of brown adipose tissue from WT male mice at 2M, 12M, and 24M of age. Ponceau's staining as loading control. **c** Quantitation of NLRP1b expression in brown adipose tissue of male mice at 2M, 12M, and 24M of age. $p = 0.0001$ (2M vs. 12M), $p < 0.0001$ (2M vs. 24M), $p < 0.0001$ (12M vs. 24M). **d** Quantitation of UCP1 expression in brown adipose tissue of male mice at 2M, 12M, and 24M of age. $n = 7$ (2M), $n = 6$ (12M), $n = 3$ (24M) mice. $p < 0.0001$ (2M vs. 12M), $p = 0.020$ (2M vs. 24M). **e** Perilipin immunofluorescence (red) and DAPI (blue) staining of nuclei in brown adipose tissue of WT male mice at 2M and 24M of age. Arrows indicate perilipin-depleted cells. Scale bars: 100 μM. **f** Quantification of perilipin-depleted cells was performed on brown adipose tissue from WT male mice at 2M and 24M of age. $n = 3$

mice. $p = 0.001$. **g** Quantification of IL-18 level by ELISA in plasma of 12M old male mice being treated for 4 months with vehicle, VX-765 (50 mg/kg, three times a week) or Z-DEVD-FMK (2.5 mg/kg, three times a week). $n = 7$ mice. $p = 0.012$ (Vehicle vs. VX-765), $p = 0.011$ (VX-765 vs. Z-DEVD-FMK). **h** Body weight of the mice treated with Vehicle, VX-765 (50 mg/kg, three times a week) or Z-DEVD-FMK. $n = 10$ (Vehicle and Z-DEVD-FMK), $n = 8$ (VX-765) mice. **i** The mass of subcutaneous adipose tissue normalized to body weight of the mice treated with Vehicle, VX-765 or Z-DEVD-FMK (2.5 mg/kg, three times a week). $n = 10$ (Vehicle and Z-DEVD-FMK), $n = 8$ (VX-765) mice. **j** The mass of brown adipose tissue normalized to body weight of WT mice treated with Vehicle, VX-765 or Z-DEVD-FMK. $n = 8$ (Vehicle and VX-765), $n = 10$ (Z-DEVD-FMK) mice. $p = 0.046$. **k** Core body temperature of WT male mice treated with Vehicle, VX-765 or Z-DEVD-FMK after acute cold challenged from thermoneutrality to 0 °C for 60 min. $n = 9$ mice. Comparison of Vehicle with VX-765, $p = 0.0056$ (10 min), $p = 0.0017$ (20 min), $p = 0.0008$ (30 min), $p = 0.013$ (40 min), $p = 0.0087$ (50 min), $p = 0.0018$ (60 min). Comparison of Vehicle with Z-DEVD-FMK, $p = 0.0489$ (10 min), $p = 0.0321$ (40 min). All data represent the mean ± SEM. *$p < 0.05$, **$p < 0.01$, ***$p < 0.001$, ****$p < 0.0001$ and #$p < 0.05$ by two-tailed Student's $t$ test (**a, f, g, k**) or one-way ANOVA (**c, d, i, j**).

---

control (Ade-CMV). Ten days post infection there was a near complete ablation of STX4 protein with a marked reduction of UCP1 protein concomitant with increased cleaved Caspase1 (Fig. 6a). Quantitative real-time PCR data demonstrated the decreased expression of *Stx4a* and *Ucp1* mRNA in the STX4KO brown adipocytes associated with the increased mRNA expression of the pyroptotic genes *Casp1, Nlrp1b* and *Il18* (Fig. 6b). Moreover, the STX4KO adipocytes displayed reduced number of nuclear positive HMGB1 staining (marker of cell death, Fig. 6c, d) and increased secretion of mature IL-18 from medium (Fig. 6e).

To determine if the adipocyte death was a result of activated pyroptosis, we differentiated STX4^{fl/fl} primary brown adipocyte progenitor cells and treated the cells with vehicle or the Caspase1/11 inhibitor (VX-765) and Caspase3 inhibitor (Z-DEVD-FMK) 3 days after infection with Ade-CMV or Ade-Cre (Fig. 6f). STX4KO brown adipocytes displayed higher Caspase1 activity in vehicle and Z-DEVD-FMK-treated cells whereas VX-765 treatment significantly reduced Caspase1 activity. In the control, Ade-CMV infected cells there were few positive propidium iodine (PI) staining cells indicating that most cells maintained their plasma membrane integrity (Fig. 6g). In contrast, following

infection with Ade-Cre there was a high percentage (~30%) of PI-positive cells. However, treatment with the Caspase1/11 inhibitor (VX-765) significantly suppressed the number of PI-positive cells to 10% (Fig. 6g, h). We also confirmed these findings by measuring the activity of lactate dehydrogenase released into the media as another indicator of cell death (Fig. 6i). To further confirm that the activation of pyroptosis through Caspase1/11, we made STX4KO brown adipocytes that were infected with lentivirus encoding a random (NMshRNA), Caspase1 (Casp1shRNA), Caspase3 (Casp3shRNA) or Caspase11 (Casp11shRNA). Immunoblotting demonstrated the effectiveness and specificity of the different shRNAs to reduce the protein levels of each target (Supplementary Fig. 5c, d). Propidium iodine (PI) labeling indicated a marked increase in the number of PI-positive (dead) cells in the STX4KO compared to wild-type (Supplementary Fig. 5e, f). Although Caspase3 knockdown had a small protective effect, the knockdown of Caspase1 and Caspase11 substantially reduced the extent of PI labeling. In addition, there was no significant difference in the release of lactate dehydrogenase activity in the Caspase3 knockdown, but both Caspase1 and Caspase11 reduced cell death almost back to wild-type levels (Supplementary Fig. 5g). These data further support that the activation

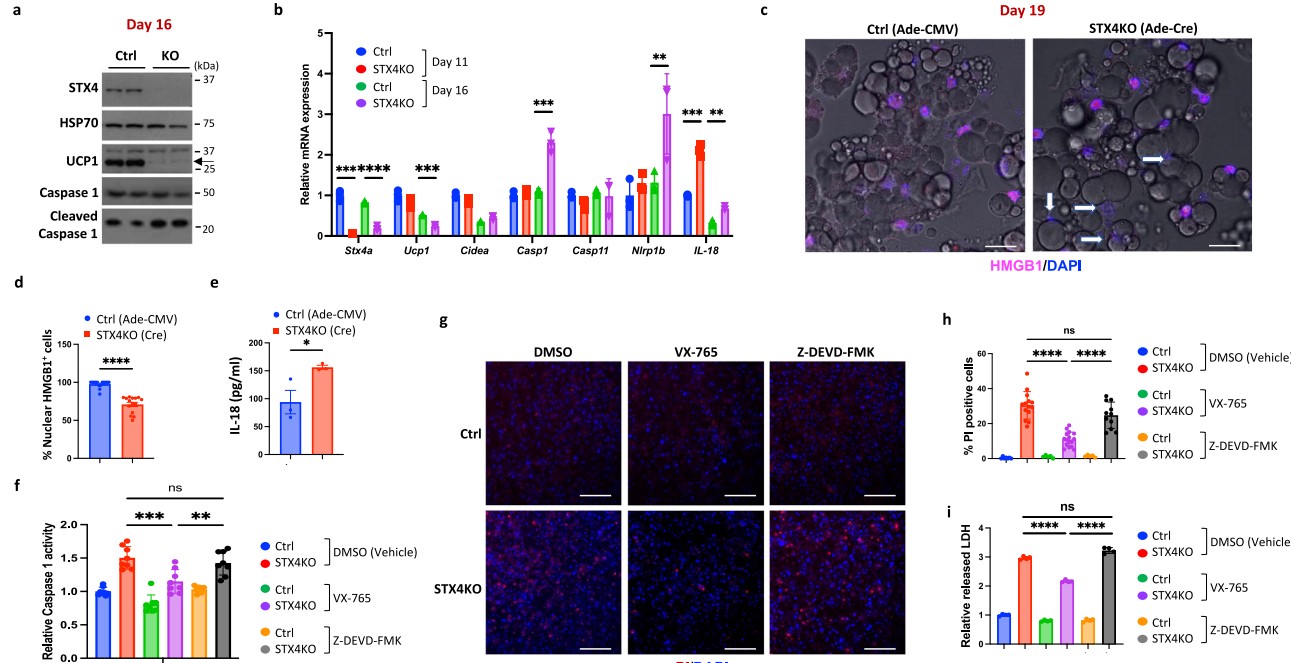

**Fig. 6 | Knockout of STX4 causes Caspase 1/11-mediated pyroptosis in a brown adipocyte autonomous manner.** Stromal vascular cells from brown adipose tissue of STX4^{fl/fl} male mice were isolated and differentiated to brown adipocytes for 5 days. Then the brown adipocytes were infected with adenovirus (Ade-CMV) as control or with an adenovirus containing Cre (Ade-Cre) to generate STX4 knockout cells. **a** Following 16 days of differentiation the control (Ctrl) and STX4KO (KO) cells were immunoblotted for the indicated proteins. **b** WT and KO brown adipocytes as above were extracted and subjected to RT-qPCR to determine the indicated mRNA levels. $n = 3$ biologically independent samples. Comparison of Ade-CMV with Ade-Cre at day 10, $p = 0.0002$ (*Stx4a*), $p = 0.0002$ (*Il18*). Comparison of Ade-CMV with Ade-Cre at day 16, $p = 7.23E\text{-}05$ (*Stx4a*), $p = 0.0002$ (*Ucp1*), $p = 0.035$ (*Cidea*), $p = 0.0014$ (*Casp1*), $p = 0.045$ (*Nlrp1b*), $p = 0.0016$ (*Il18*). **c** The representative image of HMGB1 immunofluorescence staining (red) and DAPI (blue) staining of nuclei of brown adipocytes following 19 days of differentiation. **d** Quantitation of HMGB1 staining. About 500 cells for each group are counted. $p = 2.9E\text{-}09$. **e** IL-18 ELISA from medium of primary brown adipocytes at day 16. $n = 3$ biologically

independent samples. $p = 0.043$. **f** Relative Caspase1 activity of day 16 primary adipocytes treated with VX-765 (50 μM) or Z-DEVD-FMK (50 μM) three days following infection with Ade-CMV or Ade-Cre. $n = 8$ biologically independent samples. $p = 0.001$ (STX4KO DMSO *vs.* STX4KO VX-765), $p = 0.0037$ (STX4KO VX-765 *vs.* STX4KO Z-DEVD-FMK). **g** The representative image of PI/Hoechst staining of day 19 primary adipocytes treated with VX-765 (50 μM) or Z-DEVD-FMK (50 μM) three days following infection with Ade-CMV or Ade-Cre. **h** Quantitation of the percentage of PI-positive cells. About 500 cells for each group are counted. $p < 0.0001$ (STX4KO DMSO *vs.* STX4KO VX-765), $p < 0.0001$ (STX4KO VX-765 *vs.* STX4KO Z-DEVD-FMK). **i** The released LDH level from medium of d19 primary adipocytes with VX-765 (50 μM) or Z-DEVD-FMK (50μM) three days following infection with Ade-CMV or Ade-Cre. $n = 4$ biologically independent samples. $p < 0.0001$ (STX4KO DMSO *vs.* STX4KO VX-765), $p < 0.0001$ (STX4KO VX-765 *vs.* STX4KO Z-DEVD-FMK). Scale bars: 200 μM. All data represent the mean ± SEM. ** $p < 0.01$, *** $p < 0.001$, **** $p < 0.0001$ by two-tailed Student's *t* test (**b, d, e**) or one-way ANOVA (**f, h, i**).

---

of Caspase1/11 is downstream of STX4 in a cell-autonomous manner and functionally responsible for the brown adipocyte cell death.

## STX4 functions as a regulator of energy metabolism

The pyroptosis pathway can be activated by both extrinsic (pathogens) and intrinsic signals[30,36–40]. To investigate the molecular mechanisms of STX4-regulated brown adipocyte pyroptosis, we developed an inducible brown adipocyte cell line by taking Stx4^{fl/fl} brown adipocyte progenitor cells and transfecting them with SV40 lentivirus, and retrovirus Cre^{ERT2} (Supplementary Fig. 6a). Following 6 days of brown adipocyte differentiation, the cells were treated with 4-hydroxytamoxifen (4-OHT) for 4 days, which led to a near complete loss of *Stx4a* mRNA by day 10 (Supplementary Fig. 6b) and a near-total depletion of STX4 protein by day 12 (Fig. 7a). Concurrently, there was an increase in cleaved Caspase1 protein, indicating pyroptosis activation. In parallel, the STX4KO cell line displayed increased cell death as determined by propidium iodide (PI) labeling at day 16, but not at day 12 (Fig. 7b). Therefore, we used day 12 brown adipocytes for subsequent experiments to identify the signaling mechanisms and molecular events responsible for the pyroptosis activation of cell death.

Targeted metabolic profiling revealed a clear separation of the control and STX4KO cell lines by PCA analysis (Supplementary Fig. 6c). In particular the SXT4KO cells had decreased levels of pyruvic acid, lactic acid, and 3-phosphoglyceric acid, along with increased beta-

glycerophosphate levels (Fig. 7c), that is indicative for the down-regulation of glycolysis[41]. Seashore flux analyses of the extracellular acidification rate (ECAR) was reduced in the STX4KO brown adipocytes (Fig. 7d, e). The reduction in ECAR was also consistent with a reduction in insulin and norepinephrine-stimulated glucose uptake (Fig. 7f) that is dependent upon STX4-mediated GLUT4 and GLUT1 vesicle fusion with the plasma membrane[42–46]. Consistent with the reduction in ECAR being due to decreased glucose uptake, when pyruvate was used to bypass glucose uptake, ECAR was unaffected in the Ucp1-STX4KO brown adipocytes (Supplementary Fig. 6d). In addition to the reduction in glycolysis, the STX4KO brown adipocytes also displayed reduced maximal oxygen consumption rate (OCR) and reserve oxygen consumption capacity (Fig. 7g, h) with reduced norepinephrine stimulated OCR (Fig. 7i, j). The combination of reduced glycolysis coupled with reduced mitochondria electron chain activity resulted in decreased cellular ATP levels (Fig. 7k), which has been identified as cell-intrinsic pyroptosis activation signal[36]. Importantly, there was no change in mitochondria number (Fig. 7l) demonstrating that the defect in oxygen consumption is likely due to a direct effect on mitochondrial dysfunction.

## STX4 can localize to the inner mitochondria membrane and regulates UCP1 protein stability

In addition to its plasma membrane localization, STX4 has been found to be localized to the mitochondria in skeletal muscle, where it

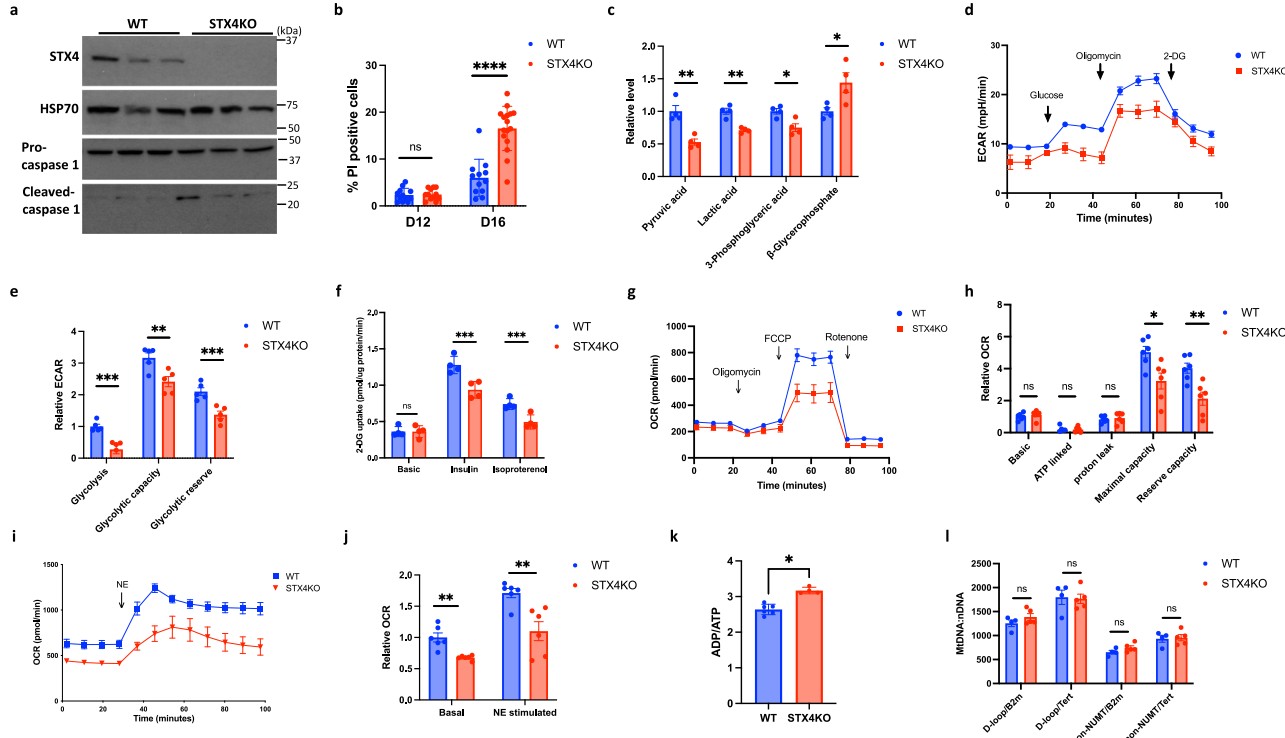

**Fig. 7 | Stx4 functions as a regulator of energy metabolism.** Brown adipocyte progenitor cells from STX4$^{fl/fl}$ male mice were transfected with SV40 lentivirus and retrovirus Cre$^{ERT2}$. Then following 6 days of brown adipocyte differentiation, the cells were treated with 1 μM 4- hydroxy-tamoxifen (4-OHT) for 4 days to knockout STX4 (STX4KO or 4OHT), while the cells treated with methanol as control (WT or methanol). **a** Representative immunoblotting of WT or STX4KO brown adipocytes at day 12 with different antibodies. HSP70 as loading control. **b** Quantitation of the percentage of PI-positive brown adipocytes at day 12 and day 16 of differentiation. About 500 cells for each group are counted. $p < 0.0001$. **c** Relative level of several metabolites related with glycolysis from WT and STX4KO brown adipocytes at day 12. $n = 4$ biologically independent samples. $p = 0.003$ (Pyruvic acid), $p = 0.001$ (Lactic acid), $p = 0.014$ (3-Phosphoglyceric acid), $p = 0.033$ (β-Glycerophosphate). **d** Extra cellular acidification rate (ECAR) in WT and STX4KO brown adipocytes was tested by Seahorse XF Analyzer, at day 11 of differentiation. **e** Quantitative analysis of the ECAR data obtained in (**d**). $n = 5$ biologically independent samples. $p = 0.0019$ (glycolysis), $p = 0.010$ (glycolytic capacity), $p = 0.002$ (glycolytic reserve). **f** Glucose uptake in WT and STX4KO brown adipocytes at day 12 in the absence or presence of 1 μM insulin and 1 μM isoproterenol for 40 min. $n = 4$ biologically independent samples. $p = 0.0061$ (Insulin), $p = 0.0069$ (Isoproterenol). **g** Oxygen consumption rate in WT and STX4KO brown adipocytes was measured by Seahorse XF Analyzer, at day 11 of differentiation. **h** Quantitative analysis of the OCR data obtained in (**g**). $n = 6$ biologically independent samples. $p = 0.015$ (Maximal capacity), $p = 0.004$ (Reserve capacity). **i** Oxygen consumption rate was measured in WT and STX4KO brown adipocytes at day 11 of differentiation treated with 1 μM norepinephrine by Seahorse XF Analyzer under 5.5 mM glucose medium. **j** Quantitative analysis of the OCR data obtained in (**i**). $n = 6$ biologically independent samples. $p = 0.0014$ (Basal), $p = 0.0046$ (NE stimulated). **k** The ratio of ADP/ATP in WT and STX4KO brown adipocytes at day 12 of differentiation was measured by the ADP/ATP Ratio assay kit. $n = 6$ (WT), $n = 4$ (STX4KO) biologically independent samples. $p = 0.017$. **l** The ratio of mitochondria DNA to nuclear DNA in WT and STX4KO brown adipocytes at day 12 of differentiation using different sets of primers. $n = 4$ (WT), $n = 5$ (STX4KO) biologically independent samples. All data represent the mean ± SEM. *$p < 0.05$, **$p < 0.01$, ***$p < 0.001$, ****$p < 0.0001$, by two-tailed Student's $t$ test.

regulates mitochondrial fission[47]. Subcellular fractionation of wild-type and STX4KO brown adipocytes revealed that although the majority of the endogenous STX4 protein is present in non-mitochondrial membranes there was a significant amount of STX4 located in the mitochondria (Fig. 8a). Moreover, protease K digestion to degrade outer mitochondria proteins (TOM70) demonstrated that STX4 is localized to the inner mitochondrial membrane as it was resistant to protease K digestion similarly to a bona fide inner mitochondrial membrane protein, COX IV (Fig. 8b and Supplementary Fig. 6e). Furthermore, we consistently observed a lower UCP1 protein level in STX4KO brown adipocytes compared to wild-type control (Figs. 6a and 8c), even before any change in *Ucp1* mRNA (Supplementary Fig. 6b). This finding prompted us to investigate whether UCP1 protein degradation is increased by STX4 deficiency that may precede mitochondria dysfunction. To address this issue, we labeled wild-type control and STX4KO cultured adipocytes with stable isotopes labeled histidine and cysteine for 2 days that were subsequently chased with unlabeled amino acids. Quantification of the heavy (H) isotope labeled UCP1 protein divided by the unlabeled light protein (L) indicated a more rapid degradation of UCP1 in the STX4KO adipocytes (Fig. 8d). Recently it had been reported that UCP1KO mice display a

reduction in electron chain proteins with impaired mitochondrial function[48]. Consistent with these findings, western blotting for several electron chain complex proteins demonstrated a substantial reduction in brown adipose tissue from UCP1KO mice (Supplementary Fig. 6f). Similarly, the same electron chain complex subunit proteins were also decreased in brown adipose tissue from STX4KO mice, suggesting a direct relationship between UCP1 and STX4 in maintaining mitochondrial function. Transmission electron microscopy of BAT mitochondria from the Ucp1-STX4KO mice clearly indicates a marked reduction in the mitochondria area and cristae density in the Ucp1-STX4KO mice (Supplementary Fig. 6g–i), similar to that reported for UCP1KO mice[48]. In parallel, we found a downregulation of STX4 protein levels in UCP1KO mice (Supplementary Fig. 7a, b). Although UCP1KO mice do not display any increase in pyroptosis signaling when maintained at normal ambient temperature (Supplementary Fig. 7c), UCP1KO brown adipocytes are more sensitive to cell death in the presence of 2-deoxyglucose to suppress glycolysis (Supplementary Fig. 7d, e).

We next isolated mitochondria from wild-type BAT that was crosslinked with or without DSP (3,3-Dithio- bis- (sulfosuccinimidyl) propionate). STX4 immunoprecipitation demonstrated the co-

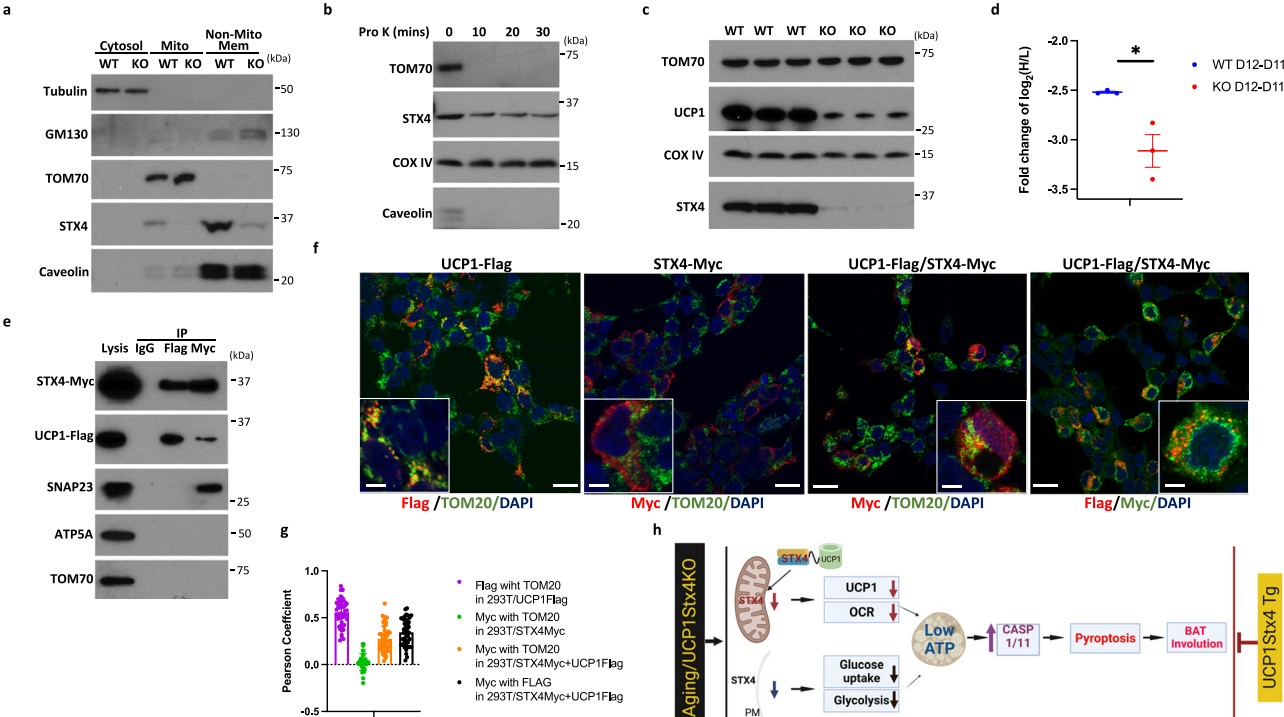

**Fig. 8 | STX4 can localize to the inner mitochondria membrane and regulates UCP1 protein stability. a** Representative immunoblotting of Tubulin, GM130, TOM70, STX4 and Caveolin in cytosol fraction, mitochondria fraction and non-mitochondrial membrane fraction from WT and STX4KO brown adipocytes at day 11 of differentiation. **b** Representative immunoblotting of TOM70, STX4, COX IV and Caveolin in mitochondria fraction from male WT (2-month-old) brown adipose tissue treated with 20 μg/μl proteinase K for 0 min, 10 min, 20 min and 30 min. **c** Representative immunoblotting of TOM70, STX4 and UCP1 in mitochondria fraction from male WT and STX4KO brown adipocytes at day 12. **d** The fold change of the log2 ratio of the heavy (H) isotope labeled UCP1 protein divided by the unlabeled light protein (L) at day 12 compared to day 11 in WT and STX4KO adipocytes. n = 3 biologically independent samples. $p = 0.023$. **e** Lysates from 293 T cell overexpressing STX4-Myc and UCP1-Flag for 24hrs were subjected to immunoprecipitation using ether Flag antibody or Myc antibody, followed by blotting with antibodies against Myc, UCP1, SNAP23, ATP5A and TOM70. **f** The 293 T cell overexpressed STX4-Myc and UCP1-Flag for 24 h were immunostaining for Flag (green) and Myc (red) with DAPI (blue) labeling of nuclei. These pictures are representative of 50 cells. Scale bar: 20 μm and 5 μm (insert). **g** The extent of co-localized STX4-Myc and UCP1-Flag was determined by Pearson's coefficient from 3 independent experiments with the quantification from 50 cells/experiment. **h** Hypothetical schematic model that accounts for the loss of STX4 mediating both mitochondrial and glycolytic dysfunction necessary for the activation of brown adipocyte pyroptosis (Created with BioRender.com). Each experiment has been repeated for 3 times. All data represent the mean ± SEM. *$p < 0.05$, ** $p < 0.01$, and ***$p < 0.001$, by two-tailed Student's $t$ test.

immunoprecipitation of UCP1 and SNAP23, but not TOM70 or COX IV (Supplementary Fig. 7f). To further confirm the interaction between STX4 and UCP1, we overexpressed STX4-Myc and UCP1-Flag in 293T cells for 24 hrs and performed immunoprecipitation and immunostaining using Flag or Myc antibodies. Immunoprecipitation of STX4-Myc resulted in the co-precipitation of UCP1-Flag and immunoprecipitation of UCP1-Flag resulted in the co-precipitation of STX4-Myc (Fig. 8e). The transfected UCP1-Flag protein co-localized with TOM20 indicating mitochondrial co-localization whereas the transfected STX4-Myc was distinct from TOM20 and appeared primarily plasma membrane localized (Fig. 8f). However, co-transfection of UCP1-Flag with STX4-Myc resulted in a significant redistribution of STX4-Myc that now co-localized with TOM20 and with UCP1-Flag. Quantification of these data using Pearson's coefficient is shown in Fig. 8g. Further, we observed a reduction in STX4 and UCP1 protein under thermoneutral conditions (Supplementary Fig. 7g–j) to demonstrate the role of STX4 in stabilizing the UCP1 under physiological conditions. Together these data indicate that STX4 interacts with UCP1 protein and is necessary to maintain UCP1 protein stability.

## Discussion

It is well established that humans are born with classical brown adipose tissue localized to the interscapular and perirenal regions[49] with the interscapular depot regressing in the first and second decade of life[17,20] and the perirenal depot slowly transforming into a white adipose tissue

depot[50]. Since thermogenic adipose tissue increases insulin sensitivity and energy expenditure to suppress the development of diabetes and obesity, there are currently numerous efforts to find therapeutic approaches to induce the formation of beige adipose tissue[51–53]. However, an alternative approach would be to prevent the regression of brown adipocytes that are present at birth. Although mice do not fully regress their brown adipose tissue, during the aging process classical brown adipocyte numbers decline with decreased thermogenic capacity[23]. Thus, understanding the post-developmental changes of thermogenic brown adipose tissue may have important therapeutic implications for the improvement of metabolic health and to improve energy balance.

In the case of white adipose tissue, numerous studies have clearly documented that these depots undergo hypertrophy and adipocyte cell death during diet-induced obesity[54–57]. Although studies have suggested that this results from apoptotic, necrotic, or autophagic induced cell death[54–57], based upon extensive morphological criteria, pyroptotic death was the primary cause in hypertrophic white adipocytes[58,59] suggesting that this may be a generalized adipocyte response to excessive lipid accumulation[60]. This is consistent with a recent study showing that genetic (ChREBP deficiency) and pharmacological inhibition of lipogenesis under thermoneutrality preserves mitochondria mass and thermogenic capacity of BAT[61].

Our data clearly demonstrate that during the aging process, pyroptosis activation is a key driver of brown adipocyte cell death and a

decline in thermogenic activity of the remaining brown adipose tissue depots. Recent studies have shown that rabbit stromal vascular cells expressing high levels of follistatin (FSTL) serve as brown adipocyte progenitors and loss of FSTL expression in mice results in decreased iBAT mass due to a suppression of progenitor cell differentiation[62]. In contrast, our snRNAseq data from iBAT of aging mice did not indicate any significant loss of brown adipocyte progenitor cells but instead the appearance of an *Ucp1*-low adipocyte population concomitant with a reduction in the *Ucp1*-high adipocyte cells in both aging and STX4KO mice. While both *Ucp1*-high and *Ucp1*-low cells in the KO mice show expression of pyroptotic genes, the levels of these genes are significantly higher in the *Ucp1*-low cells compared to the *Ucp1*-high cells. Additionally, trajectory inference analyses indicated that the *Ucp1*-low adipocytes were derived from the *Ucp1*-high adipocytes suggesting a de-differentiation process that subsequently undergoes a pyroptotic cell death program. Future work will require direct evidence such as lineage tracing to verify this finding. Although this can account for the decline in BAT mass and the reduction in UCP1 protein could account for the defect in uncoupled mitochondrial respiration, it cannot account for the defect in coupled respiration in aged mice.

In our analyses, we observed a concomitant decline in STX4 which has previously reported as a healthy longevity gene and that protects pancreatic beta cells and skeletal muscle from metabolic dysfunction[26,27,47]. We, therefore, generated both brown adipocyte-specific STX4 deficient mice, which mimic aging-induced reduction in brown adipose tissue mass and thermogenic dysfunction, whereas transgenic brown adipocyte STX4 overexpressing mice protects against the aging-induced decline. Moreover, pharmacological inhibition of the pyroptosis effector proteins Caspase1/11 protects against both aging and STX4 deficiency of thermogenic dysfunction in vivo and specific knockdown of Caspase1 and 11 but not Caspase 3 protected against cell death in cultured brown adipocytes.

In any case, these findings beg the question of how does STX4 regulates both brown adipocyte thermogenic function and pyroptotic cell death. Our data strongly support the hypothesis depicted in Fig. 8h in which STX4 plays two important roles. One is its classical role in the trafficking and fusion of glucose transporter to the plasma membrane[63,64]. As such, we observed both a decrease in glucose uptake and glycolysis. To further demonstrate that the defect in glycolysis was due to impaired glucose uptake, we used pyruvate to bypass glucose uptake. As shown, pyruvate-stimulated ECAR was unaffected in the Ucp1-STX4KO brown adipocytes. However, the partial inhibition of adipocyte glucose uptake by itself is not sufficient to cause cell death[65], likely due to compensation by mitochondria energy production. However, we also observed a parallel inhibition of oxygen utilization indicating mitochondrial electron chain dysfunction consistent with the observed decrease in electron chain protein complexes.

How is it then that STX4 disrupts the mitochondrial respiratory chain? Our finding revealed that in brown adipocytes STX4 is physically associated with UCP1 and stabilizes it as STX4 deficiency results in an accelerated degradation rate of the UCP1 protein. Consequently, in STX4-deficient brown adipocytes, UCP1 protein levels dropped significantly and became nearly undetectable before significant death was observed. Moreover, STX4 protein levels decreased in brown adipose tissue of UCP1KO mice. We have also found that the Ucp1-STX4TG mice displayed elevated UCP1 protein levels under thermoneutrality and with aging. Together, these data clearly demonstrated a co-dependence of both UCP1 and STX4 protein levels on each other.

Recently, it was demonstrated that UCP1 deficiency not only impairs electron chain uncoupling of the proton gradient, but it also induces a defect in coupled electron transport as well as a reduction in mitochondria size with reduced cristae density[48]. The mitochondria morphologic phenotype with reduced electron chain respiration

observed in the UCP1KO mice is similar to that of the STX4KO mice. These data are consistent with mitochondrial function being co-dependent upon the interaction between STX4 and UCP1 and deficiency in one also results in the deficiency of the other and thereby results in the disruption of mitochondrial respiration. Thus, the loss of STX4 protein either by aging or by genetic manipulation prevents the association of STX4 that is necessary for the stabilization of both the respiratory chain subunits as well as UCP1 itself. We therefore hypothesize that the activation of pyroptosis results from the combined defects in glycolysis and mitochondria respiration leading to reduced ATP levels, which, in turn, activate pyroptosis and ultimately contribute to brown adipose tissue involution.

Determination of the mechanisms by which UCP1 stabilizes the mitochondrial electron transport chain and the role of STX4 protein trafficking to the inner mitochondrial membrane are important mechanistic issues to understand the complex process responsible for brown adipose tissue involution. Moreover, further analyses of brown adipocyte cell autonomous pyroptosis as the basis for aging-induced decline in brown adipose tissue mass and thermogenic function will provide therapeutic potential for preventing the decline of energy expenditure and insulin sensitivity during the aging process.

## Methods

### Mice

The mice used in this study were group-housed either at 24 °C or 29 °C with a relative humidity of 30–70%, on a 12 h light/12 h dark cycle, with ad libitum access to food and water. Euthanasia was performed via carbon dioxide inhalation to ensure an overdose, with all other animal care procedures following the guidelines of the Institutional Animal Care and Use Committee at Albert Einstein College of Medicine. All mice were maintained on a C57BL/6 J background. The C57BL/6 J mice used in this study were obtained from the Jackson Laboratory (000664, JAX). The STX4$^{fl/fl}$ mice were generously provided by Dr. Adachi Roberto, while the Ucp1-Cre mice were obtained from the Jackson Laboratory (024670, JAX). The Ucp1-cre$^{ERT2}$ mice were gifted by Dr. Christian Wolfrum.

### Rosa26-*Stx4a* KI mice

To insert a cassette consisting of Ad-SA (a splicing acceptor)-loxp-tPA (triple polyA signal)-loxp-mouse *Stx4a* CDS-PolyA signal in sequence, we used the CRISPR/HDR strategy to target intron 1 of the Rosa26 locus. Microinjections to obtain zygotes and live mouse pups were performed by Albert Einstein Gene Modification Facility and Transgenic Mouse Facility.

### Stx4$^{fl/fl}$ cell line

Stromal vascular cells were isolated from 1 week or younger embryonic STX4$^{fl/fl}$ mice and cultured in DMEM medium supplemented with 15% FBS. The cells were then immortalized with SV40 lentivirus and transfected with tamoxifen-inducible Cre retrovirus. Single clones were selected and assessed for their ability to differentiate. Only clones with good differentiation were retained for further experiments. To induce Cre recombinase-mediated recombination, 1 μM of 4-hydroxytamoxifen (4-OHT) was added to the differentiated cells for two days, followed by a change of fresh medium with 1 μM of 4-OHT for another two days.

### Lentiviral transduction

MISSION lentiviral shRNA particles (106 VP/mL) for Caspase1 (TRCN0000012267), Caspase3 (TRCN0000012254), Caspase11(TRCN0000012269) and NM (nonmammalian, SHC202V) control were obtained from Millipore–Sigma. STX4$^{fl/fl}$ cell lines were then selected with G418 (250 μg/ml) after lentivirus infection (MOI = 1) and then subjected to standard adipocyte differentiation.

## Acute cold exposure

To induce acute cold exposure, mice housed at ambient temperature (25 °C) or thermoneutrality (29 °C) were transferred to 4 °C or 0 °C for 1 h, without access to food or water. Body temperatures were monitored every 10 min using TH-8 Thermometer (Physitemp Instruments, Clifton, NJ) as per the manufacturer's instructions.

## Insulin tolerance tests

To conduct insulin tolerance test, fasted mice (from 9:00 am to 2:00 pm) were intraperitoneally injected with 1.5 or 2 mU insulin per kg of lean mass. Blood glucose levels were measured at 0, 15, 30, 60, and 120 min using a glucose meter (Infinity, US Diagnostics).

## Brown adipocyte differentiation

Precursor cell line or stromal vascular cells from brown adipose tissue were cultured in DMEM/F12 medium supplemented with 15% fetal bovine serum and 0.2% primocin until confluency. The cells were then differentiated using the MesenCult™ Adipogenic Differentiation Kit (STEMCELL Technology) with some modifications. Confluent precursor cells were differentiated in MesenCult™ Basal Medium supplemented with 1/10 of the 10X Supplement, 1/100 of the 100X Glutamax, and 1 μM of Rosiglitazone for 4 days, after which fresh differentiation medium was added every 4 days. For subsequent experiments, the differentiated cells were transferred to DMEM (5 mM glucose) medium containing 10% FBS, 1 nM T3, 20 nM insulin, 1 μM Rosiglitazone, and 0.2% primocin one day before performing the Seahorse experiment.

## Mitochondria isolation

Mitochondria isolation was performed using a Mitochondria Isolation Kit from Miltenyi Biotec with some modifications. To start, $10^7$ adipocyte cells (10 cm dish) or 100 mg brown adipose tissue were washed twice with PBS, and then 1 ml of lysis buffer containing protease/phosphatase inhibitors was added. The cells were homogenized with a Dounce homogenizer for 40 strokes, and the homogenate was centrifuged at 700 g for 5 min to remove top lipids and cell debris pellets. The instructions from the kit were then followed, passing the sample through a column, and collecting the mitochondria fraction binding to beads and the flow-through. To obtain the non-mitochondrial pellet and cytosol fraction, the flow-through was centrifuged at 14,000 g for 15 min. The supernatant contains the cytosolic fraction, and the pellet contains mostly the non-mitochondrial membrane fraction.

## Mitochondria functional assays

The oxygen consumption rates (OCRs) and extracellular acidification rate (ECAR) were determined using the XF24 Extracellular Flux Analyzer (Seahorse Bioscience, MA, USA) following the manufacturer's protocols. The differentiated adipocytes were washed three times with seahorse medium supplemented with 5 mM glucose, 2 mM glutamine and 1 mM sodium pyruvate, then transferred to a temperature-controlled (37 °C) Seahorse analyzer and allowed to equilibrate for 20 min. Basal and after port injection measurements were looped 3 times. Each loop comprises a 3 min mix, 2 min wait and 3 min measure. Compounds were added by automatic pneumatic injection followed by a single assay cycle after each. The concentrations of drugs used were oligomycin (5 μM), FCCP (1 μM), rotenone (1 μM), and glucose (10 mM).

## Glucose uptake

Differentiated adipocytes were subjected to a 6-hour fast in serum-free DMEM medium containing 5 mM glucose, 2mM L-glutamine, 10 mM HEPES, 0.5% BSA, 2.4 nM insulin, and 0.2% Primocin. Following the fast, the cells were treated for 40 min in serum-free, glucose-free, and insulin-free medium. For insulin-treated samples, 1 μM insulin was added for 20 min, while 1 μM isoproterenol was added for 20 min for isoproterenol-treated samples. Subsequently, all three groups were treated with 1 mM 2-deoxy-D-glucose (2-DG) for 20 min. Then glucose uptake was determined using the Glucose Uptake Assay Kit (ab136955), and the final data was normalized to 2-DG uptake per μg protein per minute.

## RNA extraction, qPCR analysis and bulk RNA sequencing

Total mRNA was extracted by homogenizing brown adipose tissues in TRIzol, treating with chloroform, and precipitating in 70% ethanol. cDNA was made using SuperScript IV VILO Master Mix and qPCR was performed utilizing QuantStudioTM 6 Flex Real-Time PCR System (ThermoFisher Scientific), using primers supplied by the same company. After RNA extraction from BAT of 2-month-old WT and Ucp1-STX4KO male mice, with four biological replicates each, 2 μg of total RNA was used as input for polyA(+)-RNA enriched and strand-specific library preparation, which was performed by Novogene. Sequencing was carried out on an Illumina NovaSeq 6000 machine. The reads were mapped to the mm10 reference genome using STAR software. Differential expression analysis was performed using R package DESeq2, and Gene Ontology and KEGG pathways were analyzed using R package clusterProfiler.

## Protein extraction and immunoblot

Tissues were homogenized in RIPA buffer with protease and phosphatase inhibitor cocktail. Proteins were separated using SDS-PAGE and transferred to PVDF membrane (Millipore) by iBlot 2 Western Blot Transfer System. Quantification of immunoblots was performed using ImageJ (NIH). Following antibodies were used: Vinculin (Abcam,#ab18508), Caspase1 (AdipGen Life Sciences, #AG-20B-0042); Caspase1(ThermoFisher, #PA5-38100; AdipGen Life Sciences, #AG-20b-0044); Caspase3 (Cell Signaling Technology, #9661); Caspase11 (Abcam, #ab180673); Syntaxin 4 (Synaptic Systems, #110042); Tubulin (Cell Signaling Technology, #2144); Hmgb1 (Abcam, #ab67281); UCP1 (Abcam, #ab10983); Nlrp1b (Novus Biologicals, #NBP1-54899); Cleaved Nlrp1b (AdipGen Life Sciences, #AG-20B-0084); PGAM1 (Cell Signaling Technology, # 12098); HSP70 (Abcam, #abab181606); AIM2 (Abcam, #ab93015); NLRP3 (AdipGen Life Sciences, #AG-20B-0006); GAPDH (Abcam, #ab8245); GM130 (Cell Signaling Technology, #12480); Caveolin (Cell Signaling Technology, # 3238); Myc (Cell Signaling Technology, # 2278); Flag (Sigma, #F1804); SNAP23 (Abcam, #ab3340); ATP5A (Abcam, #ab151229); TOM70 (Cell Signaling Technology, #65619); COXIV (Cell Signaling Technology, #4844); TOM20 (Cell Signaling Technology, #42406); OXPHOS (Abcam, #ab110413); Perilipin 1 (Cell Signaling Technology, #3470), F4/80 (Cell Signaling Technology, #70076).

## ADP/ATP ratio

Twelve days old of differentiated adipocyte cells were grown in white wall transparent 96-well plates and the ADP/ATP were tested by following the instruction of the ADP/ATP Ratio assay kit (Sigma).

## Stable isotope labeling by amino acids in cell culture (SILAC)

Eight days differentiated adipocyte cells were grown in DMEM medium for SILAC (# 88364, ThermoFisher scientific) supplemented with 10% dialyzed FBS (# 26400044) and amino acid with isotope-labeled $^{13}C_6$$^{15}N_2$-L-Lysine.2HCL (# 88209) / $^{13}C_6$$^{15}N_4$-L-arginine.HCL (# 89990) for 48 h, then washed three times with PBS then cultured with regular DMEM medium supplemented with 10% FBS. The fold change of the log2 ratio between UCP1 protein labeled with heavy-amino acids and UCP1 protein labeled with light-amino acids was determined at day 12 in comparison to day 11.

## Widely targeted small metabolites

Twelve days differentiated adipocyte cells grown in 60 mm dish were rinsed with 0.5 ml of ice-cold ammonium acetate solution twice then added 0.6 ml ice-cold methanol. Adipocytes were then scraped and

centrifuged, and the supernatant was collected and dried using a speedvac and stored at −80 °C until further use. The samples were analyzed on an ABsciex 6500+ coupled with a Waters UPLC. Small metabolites separation was performed on the Ace PFP column and iHILIC-p column (HILICON, 160−152-0520). A pooled quality control (QC) sample was added to the sample list for the small metabolites. The QCs samples were injected six times for coefficient of variation (CV) calculation for data quality control. Metabolites with CVs lower than 30% were used for the quantification. The data was analyzed with MetaboAnalyst 5.0 web server and accepted metabolites were entered manually using HMDB number.

### Single-nuclei RNA-sequencing (snRNA-seq)
In preparation for snRNA-seq, harvested mouse interscapular brown adipose tissues from 4 male mice were minced in nuclear preparation buffer (10 mM HEPES, 1.5 mM MgCl2, 10 mM KCl, 250 mM Sucrose, 0.1% NP-40, 0.6 U/µl Rnase inhibitor), then homogenized with 5 strokes in a Dounce homogenizer, this step was optimized to specifically enrich for brown adipocyte nuclei. The homogenate was passed through a 100 µM filter and centrifuged at 750 x g for 5 min. The pellet was resuspended in 2% BSA/PBS, filtered again and sorted using a MoFloXDP Cell Sorter using the gating strategy previously described[29]. After sorting nuclei were concentrated and loaded on a Chromium Controller (10x Genomics, Pleasanton, CA). Nuclei lysis, RNA extraction, and cDNA synthesis were performed using the Single Cell 3' v3.1 Dual Index Gene Expression Reagent kit (10x Genomics) following the manufacturer's instructions. The quality and size of the cDNA library were determined by TapeStation. Sequencing of the library was performed using Illumina NovaSeq at 20,000 read pairs per cell.

### Bioinformatics (single-nuclei RNA sequencing)
FASTQ files were aligned against the mouse mm10 reference and converted to gene expression on the basis of Unique Molecular Identifier (UMI) using Cell Ranger Software (version 6.1.2) with "--force-cells 10000". Obtained single-cell data were normalized with sctransform, integrated for batch effect correction, and mapped to two-dimensional uniform manifold approximation and projection (UMAP) space using Seurat (version 4.1.0). Cellular populations were delineated by FindClusters function in Seurat with "resolution = 0.3". To remove multiples and low-quality droplets, barcodes that did not have a number of detected UMIs between 200 and 2000 and a number of detected genes between 100 and 3,000 were filtered out. Violin plots were generated using Loupe Browser (version 6.4.1).

### Immunochemistry and H&E staining of adipose tissue
Perilipin staining was performed as previously described in ref. 55. Briefly, adipose tissue was fixed for 24−36 h at room temperature in zinc-formalin fixative and embedded in paraffin. Paraffin-embedded adipose tissue was sectioned, deparaffinized, and heated in antigen unmasking solution for antigen retrieval, and then cooled, washed, and blocked with 10% goat serum in Tris-buffered saline with Tween (10 mM Tris-HCl, pH 7.5, 150 mM NaCl, 0.05% Tween-20). The sections were incubated overnight with perilipin 1 (3470, Cell Signaling Technology) or UCP1 (ab10983, Abcam) primary antibodies followed by fluorescence-conjugated secondary antibodies. The sections were counterstained in Pro-Long Gold Antifade Reagent (Invitrogen, ThermoFisher Scientific) with DAPI and visualized by confocal fluorescence microscopy. Zinc-fixed, paraffin-embedded fragments of adipose tissues were sectioned (5 µm-thick) and subjected to standard H&E staining at the Albert Einstein College of Medicine Histology Core Facility or stained with cleaved Caspase1 (pa5-38099, Invitrogen) or F4/80 (70076, Cell Signaling Technology) followed by SPlink HRP detection kit with DAB chromogen (D03-18, OriGene).

### Cell death determination
Plasma membrane permeability was determined by incubating adipocytes with 250 ng/ml PI and 5 µg/ml Hoechst 33342 for 10 min and then analyzing them under a microscope. Lysed (dead) cells, which released lactate dehydrogenase (LDH) were analyzed by CytoTox 96 Non-Radioactive Cytotoxic assay (Promega). Before analysis, adipocyte cells were cultured with DMEM medium without phenol red, supplemented with 5% FBS for 6 h, then 50 µl medium for assay by following the instruction from the kit. Dead cells were excluded by HMGB1 staining, thus d16 adipocytes were stained with Hmgb1 antibody and counterstained with DAPI.

### IL-18 ELISA
Ten- or sixteen-days old adipocyte cells were cultured with DMEM medium without phenol red, supplemented with 10% FBS for overnight, then 50 µl medium for IL-18 ELISA by following the instruction from the kit (Mouse IL-18 ELISA Kit). For plasma IL-18 ELISA, we used 50 µl plasma from each mouse.

### Plasma IL-1β ELISA
50 µl plasma from WT and TG mice were collected and subjected for IL-1β ELISA by following the instruction from the kit (Mouse IL-1βeta/il-1f2 Quantikine ELISA Kit).

### Caspase1 activity
Differentiated adipocyte cells were grown in white wall transparent 96-well plates and Caspase1 activity was tested by following the instruction of the Caspase-Glo® 1 Inflammasome Assay (Promega).

### Transmission Electron Microscopy (TEM)
BAT from 3-month-old WT and Ucp1-STX4KO mice was subjected to TEM as previously described. Briefly, the tissue was fixed in 2.5% glutaraldehyde in 100 mM sodium cacodylate, pH 7.4, and postfixed in 1% osmium tetroxide in sodium cacodylate followed by 1% uranyl acetate. After ethanol dehydration and embedment in LX112 resin (LADD Research Industries), ultrathin sections were stained with uranyl acetate followed by lead citrate. Sections were examined on JEOL 1400Plus Transmission Electron Microscope in backscatter mode using an accelerating voltage of 120 kV. The number of mitochondria and the density of cristae was counted manually.

### Quantification and statistical analyses
Prism (8.1) GraphPad Software was used for data processing, analyses, and graph productions in the experiments. To analyze the differences of body temperature at different time points among two to four groups of the mice, the tabs of Grouped Analyses/two-tailed Student' t test/ANOVA (or mixed model)/multiple comparisons were selected. The subtab 'Compare each cell mean with the other cell mean in that row' was chosen to show the 'Sidak's multiple comparison test,' which summarized the statistical test results. The number of independent experimental replications and the average with standard deviation are provided in the figure legends. Unpaired two-tailed $p$-value $t$-tests were used for the statistical tests between the two groups. The statistical analyses were made at significance levels as follows: ns, not statistically significant; *$p < 0.05$; **$p < 0.01$; ***$p < 0.001$, and ****$p < 0.0001$.

### Reporting summary
Further information on research design is available in the Nature Portfolio Reporting Summary linked to this article.

## Data availability
The data supporting the findings of this study are all available in the manuscript and its Supplementary information. The snRNA sequencing data for Fig. 1 and bulk RNA sequencing data for Fig. 3 and

Supplementary Fig. 3 are available at GSE248816; The snRNA sequencing data for Supplementary Fig. 4 are available at GSE229999.

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

## Acknowledgements

We are grateful to Dr. Adachi Roberto for providing Stx4^fl/fl mice, Christian Wolfrum for UCP1-Cre^ERT2 strain. We thank Dr. Victor Shuster for his helpful discussion. We thank Einstein Gene Modification Facility and Transgenic Mouse Facility for generating Rosa26-Stx4a KI mice, Genomics Facility for assistance in single nuclei library generation, Flow Cytometry Core for single nuclei and single cells sorting, Analytical Imaging Facility for all image analysis, Histology Core for H&E staining, and the Isotope & Metabolomics Core for metabolites and Seahorse analyses. This work was supported in part by National Institutes of Health grants DK128839, DK020541, DK110426, DK026687, AFAR 22059 and 21139, CA013330 from Albert Einstein college of medicine.

## Author contributions

X.Y. generated experimental data, assisted in the data analyses and figures preparation. G.B. generated experimental data and assisted in the data analyses. T.P.W. performed experiments and contributed to project development. S.K. performed experiments and contributed to discussion. Z.S. developed the inducible Stx4 knockdown brown adipocyte precursor cell line. L. L. performed all the bioinformatics analysis. M.Z. provided UCP1KO mice for the study. A. X. performed experiments for single cell RNA sequencing. H.W. conducted the Western Blot experiments, F. C. performed animal husbandry and mouse genotyping. S.S. did the mass spectrometry data. F.Y. provided advice and discussion of results. K.S. supervised the snRNA-seq experiment and bioinformatics, prepared figures, and edited the manuscript. J.E.P. supervised the project and edited the manuscript. D.F. conceived the ideas, performed experiments, supervised the project, and drafted the manuscript. All authors have commented on the manuscript.

## Competing interests

The authors declare no competing interests.

## Additional information

**Supplementary information** The online version contains Supplementary material available at https://doi.org/10.1038/s41467-024-46944-y.

**Peer review information** : *Nature Communications* thanks Farnaz Shamsi, Jingbo Pi and the other, anonymous, reviewer(s) for their contribution to the peer review of this work. A peer review file is available.

