## [Peer Review File · Nature Communications]

Involution of Brown Adipose Tissue through a Syntaxin 4 Dependent Pyroptosis PathwayREVIEWER COMMENTS

Reviewer #1 (Remarks to the Author):

The manuscript investigates the impact of pyroptosis on the decline in brown adipose tissue (BAT) mass and thermogenesis during aging. Additionally, it identifies syntaxin 4 (Stx4a) as a negative regulator of pyroptosis in aging-induced BAT. The research addresses a crucial question that holds significant relevance in the field of adipose biology. The conclusion that Stx4a plays a vital role in regulating aging-induced pyroptosis and BAT dysfunction is supported by compelling experimental data, including results from the Stx4a gain- and loss-of-function animal models. Despite these strengths, certain sections of the manuscript lack well-supported conclusions, requiring further investigation and clarification.

Major issues

1. The identity of Ucp1 low cells: There is a lack of clarity regarding the classification of these cells as adipocytes. The authors do not provide any evidence to support their claim that these cells are indeed adipocytes and not other cell types that might have ambient Ucp1 RNA contamination. Such contamination has been frequently observed in single-cell and single-nuclei RNA-seq data of brown adipose tissue (BAT).

To support their conclusion, the authors should demonstrate the expression levels of other adipocyte markers and endothelial cell markers in the Ucp1 low cells. Based on the heatmap in figure 1c, the Ucp1 low cells are expressing high levels of endothelial markers, which further raises questions about their identity and necessitates a comprehensive examination of multiple markers to clarify their cell type.

Additionally, the authors' conclusion that "the Ucp1-STX4KO mice there was a marked increase in the Ucp1-low adipocyte clusters with a parallel reduction in the Ucp1-high adipocyte clusters compared to WT mice" is not supported clearly in supplemental Fig. 4A and B. The heatmap in supplemental Fig. 4C does not show a significant difference in Ucp1 expression between the "Ucp1-high" and "Ucp1-low" clusters. To support their claim more effectively, the authors should quantitatively show the expression levels of Ucp1 in each cluster and provide a clearer quantification of the cluster frequency. One way to achieve this is by adding the percentages of each cluster to the graph, allowing readers to easily

interpret the distribution of Ucp1-high and Ucp1-low adipocyte clusters in Ucp1-STX4KO and WT mice.

2. The proposed mechanism: The proposed mechanism in the manuscript appears to have some gaps and inconsistencies that need to be addressed. Although in vivo evidence supports a role for STX4 in pyroptosis and aging-induced BAT dysfunction, some mechanistic aspects of the proposed model have not been directly demonstrated.

One concern is related to the claim that STX4 interacts with the UCP1 protein and is necessary for maintaining UCP1 protein stability. The in vitro overexpression studies indicate that STX4-Myc is predominantly localized in the plasma membrane, and only when UCP1 is overexpressed can STX4 be detected in the mitochondria. This raises questions about the role of STX4 in stabilizing UCP1 protein in physiological conditions. Further evidence are needed to demonstrate the direct function of STX4 in UCP1 protein stabilization. Additionally, the manuscript lacks evidence to support a direct link between STX4 and glucose uptake and glycolysis. The authors are encouraged to provide further direct evidence to bolster their mechanistic model or alternatively consider refining their conclusion while taking into account the existing evidence and any potential limitations.

Minor issues:

Figure 1D: What are the pyroptotic gene markers shown in the graphs?

The reduction of IL1B in figure 2i suggests that BAT is a major contributor to circulating IL1B. Are there any evidence in the literature that support this conclusion? If so, it would be helpful to discuss those in the manuscript.

Reviewer #2 (Remarks to the Author):

In this study, the authors aimed to investigate the mechanisms underlying aging induced decline in BAT mass and thermogenic dysfunction using multiple animal and cell models, including Ucp1-STX4KO, Ucp1-STX4TG and aged mice. In general, the findings are interesting

and important. However, the major conclusions cannot be fully supported by their findings. The following major and minor points should be clarified or addressed before further consideration.

Major points:

1. Title: Given that most of the experiments were conducted using Ucp1-STX4KO and Ucp1-STX4TG mice under various conditions, including at thermoneutrality, the current title “Brown Adipocyte Pyroptosis Mediates the Aging Induced Decline in Brown Adipose Tissue Mass and Thermogenic Dysfunction” does not fit well with the key findings. Although the authors provide evidence that brown adipocyte STX4 deficiency mimics many of the involution characteristics that occur during aging, thermoneutrality and chronic high fat diet feeding, Ucp1-STX4KO mice are not aged model. The conclusions from STX4-manipulated mice/cells cannot be simply extrapolated to aging.
2. Figure 1: Why are the relative percentage of adipocyte subclusters so low, even in 2 month mice, whereas the relative percentage of endothelial cells (subcluster I-V) are so high?
3. Figure 4-6 and supplemental Fig. 5A: The caspase 1/11 inhibitor (VX-765) was used in the studies to support the major conclusion that the activation of caspase 1/11 is the key mediating the involution of BAT in STX4KO mice and with aging. While the protective effects of pharmacological caspase 1/11 inhibition (VX-765) are significant, the differences from Vehicle or Z-DEVD-FMK-treated group are somehow limited. More powerful and specific genetic manipulation approaches, such as knockout or knockdown, are highly recommended to have more solid conclusions.
4. Aging or Ucp1-STX4KO may result in many biological consequences in BAT, including BAC pyroptosis. The caspase 1/11 is the downstream effector of pyroptosis signaling, thus the BAC pyroptosis could be partially blocked by the caspase 1/11 inhibitor (VX-765). Are there any biological effects in BAT induced by aging or Ucp1-STX4KO that cannot be rescued or prevented by the caspase 1/11 inhibitor (VX-765)?
5. While the reduction in UCP1 protein in Ucp1-STX4KO mice could account for the defect in uncoupled mitochondrial respiration, it is unclear how the UCP1 protein reduction accounts for the BAC pyroptosis and BAT involution in mice.
6. Figure 7L: The authors have summarized that “classic mouse BAT undergoes an age-

dependent degeneration process termed involution which is characterized by increased lipid deposition, increased size of lipid droplets, decreased mitochondria mass with reduction of thermogenic specific genes expression, and reduction in thermogenic activity". It is unclear how to explain the result in Figure 7L showing there was no change in mitochondria number? In addition, mitochondrial morphology analysis may provide more valuable information.

Minor points:

7. Figure 1 and other relevant figures: the label text on the figures is too small to read.

8. Supplemental Figure 2E-J: the measurements should also be normalized to the length of tibia.

Reviewer #3 (Remarks to the Author):

This manuscript investigates a role for brown adipocyte pyroptosis to mediate the age-induced decline in BAT mass and thermogenic dysfunction. The authors performed sn-RNA seq in aged mice and identified a specific brown adipocyte population of Ucp1-low cells that were pyroptotic and had reduced expression of the longevity gene Stx4a. Ucp1-STX4KO mice displayed loss of BAT mass and impaired thermogenic function, as well as increased pyroptosis, while restoration of STX4 expression and suppression of pyroptosis activation preserved BAT mass and thermogenic activity. The mechanism for this was linked to reduced oxidative phosphorylation, glucose uptake and glycolysis.

This manuscript is timely and well written and elegantly used multiple genetic mouse models. I have some comments that might strengthen / clarify some things in the manuscript.

It is not clear what sex mice are used throughout the experiments or if there are there sex differences in snRNA-seq or Stx4 expression in BAT (especially w respect to age?)

Which cells are responsible for the decreased Stx4a? Are they the mature brown adipocytes or a specific cell population within the SVF?

Does STX4-Tg protect against the affects off age-induced involution? That seems an essential question to address.

While not directly related to the questions proposed, it would be of interest to mention if there were any changes in the scWAT in the Ucp1-STX4KO and STX4-Tg mice.

Were there any differences in body weight in the Ucp1-STX4KO mice? Either with age or compared to WT?

The data investigating the interaction of STX4 with UCP1 is interesting – is STX4 altered in UCP1-/- mice (or is the pyroptosis pathway altered in UCP1-/- mice)?

Reviewer #1 (Remarks to the Author):

The manuscript investigates the impact of pyroptosis on the decline in brown adipose tissue (BAT) mass and thermogenesis during aging. Additionally, it identifies syntaxin 4 (Stx4a) as a negative regulator of pyroptosis in aging-induced BAT. The research addresses a crucial question that holds significant relevance in the field of adipose biology. The conclusion that Stx4a plays a vital role in regulating aging-induced pyroptosis and BAT dysfunction is supported by compelling experimental data, including results from the Stx4a gain- and loss-of-function animal models. Despite these strengths, certain sections of the manuscript lack well-supported conclusions, requiring further investigation and clarification.

Major issues

1. The identity of Ucp1 low cells: There is a lack of clarity regarding the classification of these cells as adipocytes. The authors do not provide any evidence to support their claim that these cells are indeed adipocytes and not other cell types that might have ambient Ucp1 RNA contamination. Such contamination has been frequently observed in single-cell and single-nuclei RNA-seq data of brown adipose tissue (BAT).

To support their conclusion, the authors should demonstrate the expression levels of other adipocyte markers and endothelial cell markers in the Ucp1 low cells. Based on the heatmap in figure 1c, the Ucp1 low cells are expressing high levels of endothelial markers, which further raises questions about their identity and necessitates a comprehensive examination of multiple markers to clarify their cell type.

After re-examining original Figure 1, we fully agree with the reviewer that this dataset was not robust and underrepresent the adipocyte cell populations. We therefore repeated the snRNAseq analyses using an optimized nuclei isolation protocol that greatly enriches for brown adipocyte nuclei. These new data are now presented with a focus on the UCP1-low and UCP-1 high adipocytes population for a better comparison with supplement Figure 4 shown for the Ucp1-STX4KO mice. The UMAP plots are now shown for the adipocyte clusters (upper UMAP plots – new Figure 1A) and with the inclusion of stromal cells (lower UMAP plots – new Figure 1A). In addition, we added a series of brown adipocyte marker genes in heatmap (Figure 1C) to confirm that they are indeed adipocytes.

We especially wish to thank the reviewer for catching the weakness of our original data presented in Figure 1 that we obviously missed, and for giving us the opportunity to provide a clearer and robust dataset in the revised manuscript.

Additionally, the authors' conclusion that “the Ucp1-STX4KO mice there was a marked increase in the Ucp1-low adipocyte clusters with a parallel reduction in the Ucp1-high adipocyte clusters compared to WT mice” is not supported clearly in supplemental Fig. 4A and B. The heatmap in supplemental Fig. 4C does not show a significant difference in Ucp1 expression between the "Ucp1-high" and "Ucp1-low" clusters. To support their claim more effectively, the authors should quantitatively show the expression levels of Ucp1 in each cluster and provide a clearer quantification of the cluster frequency. One way to achieve this is by adding the percentages of each cluster to the graph, allowing readers to easily interpret the distribution of Ucp1-high and Ucp1-low adipocyte clusters in Ucp1-STX4KO and WT mice.

As requested, the dataset in Supplement Figure 4 now includes the percentages of each cluster that allows for a direct comparison of the UCP1-high and UCP1-low adipocyte clusters in UCP1-STX4KO and WT mice.

2. The proposed mechanism: The proposed mechanism in the manuscript appears to have some gaps and inconsistencies that need to be addressed. Although in vivo evidence supports a role for STX4 in pyroptosis and aging-induced BAT dysfunction, some mechanistic aspects of the proposed model have not been directly demonstrated.

One concern is related to the claim that STX4 interacts with the UCP1 protein and is necessary for maintaining UCP1 protein stability. The in vitro overexpression studies indicate that STX4-Myc is predominantly localized in the plasma membrane, and only when UCP1 is overexpressed can STX4 be detected in the mitochondria. This raises questions about the role of STX4 in stabilizing UCP1 protein in physiological conditions. Further evidence are needed to demonstrate the direct function of STX4 in UCP1 protein stabilization.

To address the reviewer's question regarding the role of STX4 in stabilizing the UCP1 protein under physiological conditions, we have now observed a reduction in STX4 and UCP1 protein under thermoneutral conditions (New Supplemental Figure 7G-J). We have also found that the UCP1-STX4TG mice displayed elevated UCP1 protein levels under thermoneutrality (Figure 2F) and with aging (New Supplemental Figure 2M-N). Moreover, STX4 protein levels decreased in brown adipose tissue of UCP1KO mice (New Supplemental Figure 7A and B). Together, these new data with our previous co-immunoprecipitation, trafficking and pulse-chase degradation data clearly demonstrated a co-dependence of both UCP1 and STX4 protein levels on each other.

Additionally, the manuscript lacks evidence to support a direct link between STX4 and glucose uptake and glycolysis. The authors are encouraged to provide further direct evidence to bolster their mechanistic model or alternatively consider refining their conclusion while taking into account the existing evidence and any potential limitations.

Previous published studies have reported that STX4 plays an important role in regulating GLUT4-dependent glucose uptake(1-3). Using targeted metabolic analyses, we observed that UCP1-STX4KO brown adipocytes display reduced levels of glycolytic intermediates (Figure 7C) with decreased insulin and norepinephrine stimulated glucose uptake (Figure 7F). Seahorse flux analyses based upon ECAR (extracellular acidification rate) also indicated reduced glycolysis (Figure 7D and E). To further demonstrate that this was due to impaired glucose uptake, we now determined ECAR using pyruvate to bypass glucose uptake. As shown in New Supplemental Figure 6D, pyruvate stimulated ECAR was unaffected in the Ucp1-STX4KO brown adipocytes. These additional new supportive data as well as additional discussion of caveats and limitations of our data are now included in the text of the revised manuscript.

Minor issues:

Figure 1D: What are the pyroptotic gene markers shown in the graphs?

The pyroptotic gene markers are *Caspase 1*, *Caspase 11*, *Il18*, *Nlrp1b* and *Nlrp3*, these have also been added in the figure 1D.

The reduction of IL1B in figure 2i suggests that BAT is a major contributor to circulating IL1B. Are there any evidence in the literature that support this conclusion? If so, it would be helpful to discuss those in the manuscript.

Our understanding of the literature indicates that IL-1beta is primarily secreted by immune cells (ie: macrophages) and white adipocytes. However, there is a study showing that IL-1beta levels increase in the vasculature of UCP1-KO mice (4). In any case, in Figure 2i the mice were maintained at thermoneutral environment for two weeks and we speculate that involution of the BAT likely induces macrophage infiltration as the source of IL-1beta.

Reviewer #2 (Remarks to the Author):

In this study, the authors aimed to investigate the mechanisms underlying aging induced decline in BAT mass and thermogenic dysfunction using multiple animal and cell models, including Ucp1-STX4KO, Ucp1-STX4TG and aged mice. In general, the findings are interesting and important. However, the major conclusions cannot be fully supported by their findings. The following major and minor points should be clarified or addressed before further consideration.

Major points:

1. Title: Given that most of the experiments were conducted using Ucp1-STX4KO and Ucp1-STX4TG mice under various conditions, including at thermoneutrality, the current title “Brown Adipocyte Pyroptosis Mediates the Aging Induced Decline in Brown Adipose Tissue Mass and Thermogenic Dysfunction” does not fit well with the key findings. Although the authors provide evidence that brown adipocyte STX4 deficiency mimics many of the involution characteristics that occur during aging, thermoneutrality and chronic high fat diet feeding, Ucp1-STX4KO mice are not aged model. The conclusions from STX4-manipulated mice/cells cannot be simply extrapolated to aging.

We thank the reviewer for pointing this out and we have now revised the title to “Involution of Brown Adipose tissue through a Syntaxin 4 dependent Pyroptosis Pathway’ that more appropriately reflects the key findings of this study.

2. Figure 1: Why are the relative percentage of adipocyte subclusters so low, even in 2 month mice, whereas the relative percentage of endothelial cells (subcluster I-V) are so high?

As indicated in our response to reviewer 01, we now provide a more robust single nuclei RNAseq dataset of BAT from 2 months, 12 months and 24 months of age (New Figure 1). We

also thank this reviewer in recognizing that the quality of the original data was not sufficient for publication and allowing us the opportunity to correct this deficiency.

3. Figure 4-6 and supplemental Fig. 5A: The caspase 1/11 inhibitor (VX-765) was used in the studies to support the major conclusion that the activation of caspase 1/11 is the key mediating the involution of BAT in STX4KO mice and with aging. While the protective effects of pharmacological caspase 1/11 inhibition (VX-765) are significant, the differences from Vehicle or Z-DEVD-FMK-treated group are somehow limited. More powerful and specific genetic manipulation approaches, such as knockout or knockdown, are highly recommended to have more solid conclusions.

As suggested, we have now examined the effect of non-specific, caspase 3 (control), caspase 1 and caspase 11 knockdowns in both wildtype and STX4KO brown adipocytes. As shown in New supplemental Figure 5C-G), knockdown of Caspase 1 or 11 dramatically protected against adipocyte cell death.

4. Aging or Ucp1-STX4KO may result in many biological consequences in BAT, including BAC pyroptosis. The caspase 1/11 is the downstream effector of pyroptosis signaling, thus the BAC pyroptosis could be partially blocked by the caspase 1/11 inhibitor (VX-765). Are there any biological effects in BAT induced by aging or Ucp1-STX4KO that cannot be rescued or prevented by the caspase 1/11 inhibitor (VX-765)?

Lipofuscin accumulates in both aged BAT as well as in the UCP1-STX4KO mice. However, neither the caspase 1/11 inhibitor (VX-765) or the caspase 3 inhibitor (Z-DEVD-FMK) has any effect on the levels of lipofuscin. This is now indicated in the Discussion section from line 580 to 585.

5. While the reduction in UCP1 protein in Ucp1-STX4KO mice could account for the defect in uncoupled mitochondrial respiration, it is unclear how the UCP1 protein reduction accounts for the BAC pyroptosis and BAT involution in mice.

Our data indicates that UCP1KO mice do not exhibit pyroptosis or cell death under normal ambient animal care facility conditions. However, brown adipocytes deficient in UCP1 are more sensitive to cell death under fasting with low glucose or when treated with 2-deoxyglucose to inhibit glycolysis (Supplemental Figure 7D-E). Spiegelman's laboratory also reported that upon cold induced stress the UCP1KO mice upregulate the expression of Caspase 11 protein, suggesting the activation of pyroptosis (5). In our BAT STX4KO mice we also observed both a decrease in UCP1 and reduction of glycolysis. Our explanation and our data support a model that when both compromised mitochondrial respiration occurs coincident with reduced glycolysis, the cells are unable to compensate with an ensuing drop in ATP levels and activation of pyroptosis.

6. Figure 7L: The authors have summarized that "classic mouse BAT undergoes an age-dependent degeneration process termed involution which is characterized by increased lipid deposition, increased size of lipid droplets, decreased mitochondria mass with reduction of thermogenic specific genes expression, and reduction in thermogenic activity". It is unclear

how to explain the result in Figure 7L showing there was no change in mitochondria number? In addition, mitochondrial morphology analysis may provide more valuable information.

As suggested, we now include TEM data of BAT mitochondria morphology from the Ucp1-STX4KO mice at 3 months of age (New Supplemental Figure 6G-I). These data clearly indicate a marked reduction in the mitochondria area and cristae density in the Ucp1-STX4KO mice. The data presented in Figure 7L is from brown adipocytes in culture for 12 days. There is no change in mitochondria number at this time point suggesting reduced mitochondria number is not the cause of pyroptosis.

Minor points:

7. Figure 1 and other relevant figures: the label text on the figures is too small to read.

The labeling of all Figures has been corrected throughout the manuscript.

8. Supplemental Figure 2E-J: the measurements should also be normalized to the length of tibia.

We maintained STX4-TG and wildtype mice under thermoneutrality for 1 month and measured the length of tibia and found no difference (see below). As such, renormalizing the data for tibia length will not change any of the results and as far as we are aware using tibia length as a normalization factor is rather unusual and not standard in the literature.

Reviewer #3 (Remarks to the Author):

This manuscript investigates a role for brown adipocyte pyroptosis to mediate the age-induced decline in BAT mass and thermogenic dysfunction. The authors performed sn-RNA seq in aged mice and identified a specific brown adipocyte population of Ucp1-low cells that were pyroptotic and had reduced expression of the longevity gene Stx4a. Ucp1-STX4KO mice displayed loss of BAT mass and impaired thermogenic function, as well as increased pyroptosis, while restoration of STX4 expression and suppression of pyroptosis activation preserved BAT mass and thermogenic activity. The mechanism for this was linked to reduced oxidative phosphorylation, glucose uptake and glycolysis.

This manuscript is timely and well written and elegantly used multiple genetic mouse models. I have some comments that might strengthen / clarify some things in the manuscript.

It is not clear what sex mice are used throughout the experiments or if there are there sex differences in snRNA-seq or Stx4 expression in BAT (especially w respect to age?)

The data presented was predominantly from male mice, although we have provided several important aspects of these data in both genders. This is now explicitly stated in each figure legend. Specifically, there was a clear decline in STX4 protein levels in the 17-month-old female mice compared to 1-month-old female mice (New Supplemental Figure 1H-J) consistent with the findings in male mice.

Which cells are responsible for the decreased Stx4a? Are they the mature brown adipocytes or a specific cell population within the SVF?

Supplemental Figure 1C and 1D directly shows that, Stx4a mRNA levels are reduced in aging mature brown adipocytes, but not in the cell types comprising the SVF.

Does STX4-Tg protect against the affects off age-induced involution? That seems an essential question to address.

Although it would take 1-2 years to generate sufficient mice to determine the effect of STX4-TG on aged-induced involution, we had 17-month-old STX4-TG female mice BAT tissue that demonstrated an increase in BAT UCP1 levels (New Supplemental Figure 2M and N). To emulate accelerated aging, we subjected male mice to thermoneutral conditions for up to of 8-month to induce involution. Under these conditions, our STX4-TG mice exhibited elevated UCP1 protein levels and reduced pyroptotic activity, in contrast to their wild-type counterparts, as detailed in Figure 2E-J.

While not directly related to the questions proposed, it would be of interest to mention if there were any changes in the scWAT in the Ucp1-STX4KO and STX4-Tg mice.

The weight of the subcutaneous white adipose tissue (scWAT) in the Ucp1-STX4KO mice was not significant differences from wild-type (WT) littermates at 5-months of age (New Supplemental Figure 3C). In the case of the STX4-TG mice, there was no change in scWAT weight when maintained for 2 weeks at thermoneutrality. However, there was a significant reduction in scWAT following 8 months under thermoneutral conditions (New Supplemental Figure 2L).

Were there any differences in body weight in the Ucp1-STX4KO mice? Either with age or compared to WT?

There were no significant differences in body weight when compared to their wild-type (WT) littermates across the age range of 2 to 9 months and the data for 5-month-old male mice are shown in New Supplemental Figure 3B.

The data investigating the interaction of STX4 with UCP1 is interesting – is STX4 altered in UCP1^{-/-} mice (or is the pyroptosis pathway altered in UCP1^{-/-} mice)?

We found a downregulation of STX4 protein levels in UCP1^{-/-} mice, as shown in New Supplemental Figures 7A and B. As indicated in response to reviewer 02, the UCP1KO mice

do not display any increase in pyroptosis signaling when maintained at normal ambient temperature (New Supplemental Figure 7C). However, the Spiegelman's group reported that following cold stress BAT of the UCP1KO mice induce caspase 11 protein (5). We have also found that the UCP1KO brown adipocytes are more sensitive to cell death in the presence of 2-deoxyglucose to suppress glycolysis (New Supplemental Figure 7D and E). Together these data support a model that the concomitant reduction of both mitochondria respiration and glycolysis are required for activation of pyroptosis in brown adipocytes.

1. Olson, A. L., Knight, J. B., and Pessin, J. E. (1997) Syntaxin 4, VAMP2, and/or VAMP3/cellubrevin are functional target membrane and vesicle SNAP receptors for insulin-stimulated GLUT4 translocation in adipocytes. *Mol Cell Biol* **17**, 2425-2435
2. Spurlin, B. A., Park, S. Y., Nevins, A. K., Kim, J. K., and Thurmond, D. C. (2004) Syntaxin 4 transgenic mice exhibit enhanced insulin-mediated glucose uptake in skeletal muscle. *Diabetes* **53**, 2223-2231
3. Yang, C., Coker, K. J., Kim, J. K., Mora, S., Thurmond, D. C., Davis, A. C., Yang, B., Williamson, R. A., Shulman, G. I., and Pessin, J. E. (2001) Syntaxin 4 heterozygous knockout mice develop muscle insulin resistance. *J Clin Invest* **107**, 1311-1318
4. Gu, P., Hui, X., Zheng, Q., Gao, Y., Jin, L., Jiang, W., Zhou, C., Liu, T., Huang, Y., Liu, Q., Nie, T., Wang, Y., Wang, Y., Zhao, J., and Xu, A. (2021) Mitochondrial uncoupling protein 1 antagonizes atherosclerosis by blocking NLRP3 inflammasome-dependent interleukin-1beta production. *Sci Adv* **7**, eabl4024
5. Kazak, L., Chouchani, E. T., Stavrovskaya, I. G., Lu, G. Z., Jedrychowski, M. P., Egan, D. F., Kumari, M., Kong, X., Erickson, B. K., Szpyt, J., Rosen, E. D., Murphy, M. P., Kristal, B. S., Gygi, S. P., and Spiegelman, B. M. (2017) UCP1 deficiency causes brown fat respiratory chain depletion and sensitizes mitochondria to calcium overload-induced dysfunction. *Proc Natl Acad Sci U S A* **114**, 7981-7986

REVIEWER COMMENTS

Reviewer #1 (Remarks to the Author):

The authors have adequately addressed the majority of the critiques, and the revised manuscript has improved. There are only a few minor concerns that need attention.

1. The results presented in Figure 1 still do not support the conclusion that the Ucp1-low (1) cells are adipocytes. The authors should examine and present the expression of specific markers for other cell types (endothelial cells, adipocyte progenitors, etc.) to rule out the possibility that this cluster is a technical artifact resulting from low-quality nuclei or doublets.
2. In Supplemental Figure 5E, the labels corresponding to different markers (PI and DAPI) are missing.
3. The discussion of the effects of aging and caspase 1/11 inhibition on lipofuscin accumulation in BAT is presented without providing any supporting data. The authors should consider including relevant evidence if it contributes to the conclusions of the current study.

Reviewer #2 (Remarks to the Author):

The authors state that they have repeated the snRNAseq analyses using an optimized nuclei isolation protocol that greatly enriches for brown adipocyte nuclei. While the new dataset presented much higher adipocyte populations than those shown in the first submission, the data interpretation and major conclusions are confusing.

Major points:

Figure 1: the figure legend “The involution of brown adipocytes with aging is associated with the increase of pyroptotic Ucp1low population”, which is also the major conclusion made by the authors or the data interpretation, is questionable. It could be argued that “The involution of brown adipocytes with aging is associated with the decrease of Ucp1high

population". However, unfortunately, the characterization of the Ucp1^{high} population and comparison of the populations among the three age groups are missing. Are there specific brown adipocyte population(s) of Ucp1^{high} cells that were pyroptotic and/or display alterations in Stx4a expression in aged mice?

Supplemental Figure 4: Genetic ablation of STX4 in iBAT not only resulted in the presence of pyroptotic Ucp1-low brown adipocytes, but also induced a dramatic reduction in Ucp1-high brown adipocytes (C). It is unclear how to interpret the huge difference in the Ucp1-high cell population between the genotypes. In addition, as shown in D, the relative expression of pyroptotic gene markers in the Ucp1-high cells of KO mice are also much higher than those in WT mice. The biological significance of the difference also needs a clarification.

Reviewer #3 (Remarks to the Author):

Comments have all been addressed and the manuscript strengthened.

Reviewer #1 (Remarks to the Author):

The authors have adequately addressed the majority of the critiques, and the revised manuscript has improved. There are only a few minor concerns that need attention.

1. The results presented in Figure 1 still do not support the conclusion that the Ucp1-low (1) cells are adipocytes. The authors should examine and present the expression of specific markers for other cell types (endothelial cells, adipocyte progenitors, etc.) to rule out the possibility that this cluster is a technical artifact resulting from low-quality nuclei or doublets.

We appreciate the reviewer's insightful observations regarding the limitations in our data depicted in Figure 1. Indeed, the Ucp1-low (1) cells exhibit characteristics of adipocytes, as evidenced by the expression of adipocyte markers. However, this cluster also displays expression of certain immune cell markers, as indicated in the heatmap provided below. Therefore, due to the ambiguity surrounding the categorization of the Ucp1-low (1) cluster and that these represent a very small population of cells, we decided to reclassify this group of cells as an indeterminate cluster. This cluster has now been removed from analysis in Figure 1B, C, D and E. Henceforth, the cells previously designated as UCP1low (2) cells now be referred to simply as Ucp1low cells.

2. In Supplemental Figure 5E, the labels corresponding to different markers (PI and DAPI) are missing.

We thank the reviewer for pointing this out and PI/DAPI is added.

3. The discussion of the effects of aging and caspase 1/11 inhibition on lipofuscin accumulation in BAT is presented without providing any supporting data. The authors should consider including relevant evidence if it contributes to the conclusions of the current study.

Lipofuscin accumulates in both aged BAT as well as in the UCP1-STX4KO mice. However, neither the caspase 1/11 inhibitor (VX-765) or the caspase 3 inhibitor (Z-DEVD-FMK) has any effect on the levels of lipofuscin. That indicate that lipofuscin is not direct cause to lead brown adipocyte death in our mouse models, we added this in our discussion section in response to a query raised by another reviewer. However, considering that this aspect does not significantly contribute to the conclusions of our study, we have now removed this from the discussion.

Reviewer #2 (Remarks to the Author):

The authors state that they have repeated the snRNAseq analyses using an optimized nuclei isolation protocol that greatly enriches for brown adipocyte nuclei. While the new dataset presented much higher adipocyte populations than those shown in the first submission, the data interpretation and major conclusions are confusing.

Major points:

Figure 1: the figure legend “The involution of brown adipocytes with aging is associated with the increase of pyroptotic Ucp1low population”, which is also the major conclusion made by the authors or the data interpretation, is questionable. It could be argued that “The involution of brown adipocytes with aging is associated with the decrease of Ucp1high population”. However, unfortunately, the characterization of the Ucp1high population and comparison of the populations among the three age groups are missing. Are there specific brown adipocyte population(s) of Ucp1high cells that were pyroptotic and/or display alterations in Stx4a expression in aged mice?

We greatly thank the reviewer for pointing this out, now we added the expression of pyroptotic genes in Ucp1high population in three ages group in Figure 1D. The data showed that there is much lower level of pyroptotic genes in Ucp1high population compared to Ucp1low population. According to our data, aged mice and our STX4 brown adipocyte specific knockout mice had increased Ucp1low population compared to young and their WT littermates. Conversely, there was a reduction in the number of cells with Ucp1high population. While it is correct that there is a decrease in the number of Ucp1high adipocytes, only the Ucp1low population expresses higher levels of pyroptotic genes. Additionally, velocity trajectory analyses have indicated that these Ucp1low cells are derived from the Ucp1high cells. These findings align more closely

with the conclusion that the Ucp1^{low} cells are undergoing involution. We agree that at present we lack direct evidence to confirm this which will require lineage tracing in mice that is beyond the scope of this study. We appreciate the significance of this question and have now incorporated it into the discussion section (line 557-562).

Supplemental Figure 4: Genetic ablation of STX4 in iBAT not only resulted in the presence of pyroptotic Ucp1-low brown adipocytes, but also induced a dramatic reduction in Ucp1-high brown adipocytes (C). It is unclear how to interpret the huge difference in the Ucp1-high cell population between the genotypes. In addition, as shown in D, the relative expression of pyroptotic gene markers in the Ucp1-high cells of KO mice are also much higher than those in WT mice. The biological significance of the difference also needs a clarification.

We appreciate the reviewer for highlighting this point. As shown in supplemental Figure 4C, WT mice exhibited 2% Ucp1^{low} cells, while KO mice showed 39%. Conversely, Ucp1^{high} in WT is 98% while 61% in KO. Supplemental Figure 4D reveals that, Ucp1^{low} cells have a markedly higher expression of pyroptotic genes in both KO and WT group, while there is a relative increase in these genes in the Ucp1^{high} cells of KO mice compared to WT mice. This suggests that Ucp1^{high} cells might be converting into Ucp1^{low} cells that are more pyroptotic, which is further supported by our velocity analyses. In addition, these findings are consistent with that of the aged mice shown in Figure 1. In any case, the differences between genotypes are likely due to the reduction of STX4, which is about 2-fold in aging (shown in Supplemental Figures 1F and G) and completely absent in KO mice, potentially accelerating this cell converting process. The reviewer's perceptive question has enriched the discussion in our study, we have now added this in the discussion section (line 557-562).

Reviewer #3 (Remarks to the Author):

Comments have all been addressed and the manuscript strengthened.

Thank the reviewer for recognition of our work.

REVIEWERS' COMMENTS

Reviewer #1 (Remarks to the Author):

The authors have adequately addressed the critiques, and the revised manuscript has improved.

Reviewer #2 (Remarks to the Author):

My concerns have been reasonably addressed.

Reviewer #1 (Remarks to the Author):

The authors have adequately addressed the critiques, and the revised manuscript has improved.

Thank the reviewer for recognition of our work.

Reviewer #2 (Remarks to the Author):

My concerns have been reasonably addressed.

Thank the reviewer for recognition of our work.